



# A comparison of the impact of TROPOMI and OMI tropospheric NO₂ on global chemical data assimilation

Takashi Sekiya[1], Kazuyuki Miyazaki[2], Henk Eskes[3], Kengo Sudo[4,1], Masayuki Takigawa[1], and Yugo Kanaya[1]

[1]Japan Agency for Marine-Earth Science and Technology, Yokohama, Japan
[2]Jet Propulsion Laboratory/California Institute for Technology, Pasadena, CA, USA
[3]Royal Netherlands Meteorological Institute (KNMI), De Bilt, the Netherlands
[4]Graduate School of Environmental Studies, Nagoya University, Nagoya, Japan

**Correspondence:** Takashi Sekiya (tsekiya@jamstec.go.jp)

**Abstract.** This study gives a systematic comparison of the Tropospheric Monitoring Instrument (TROPOMI) version 1.2 and Ozone Monitoring Instrument (OMI) QA4ECV tropospheric NO₂ column through global chemical data assimilation (DA) integration for the period April–May 2018. DA performance is controlled by measurement sensitivities, retrieval errors, and coverage. The smaller mean relative observation errors by 16% in TROPOMI than OMI over 60°N–60°S during April–May

2018 led to larger reductions in the global root mean square error (RMSE) against the assimilated NO₂ measurements in TROPOMI DA (by 54%) than in OMI DA (by 38%). Agreements against the independent surface, aircraft-campaign, and ozonesonde observation data were also improved by TROPOMI DA compared to the control model simulation (by 12–84% for NO₂ and by 7–40% for ozone), which were more obvious than those by OMI DA for many cases (by 2–70% for NO₂ and by 1–22% for ozone). The estimated global total NOₓ emissions were 15% lower in TROPOMI DA, with 2–23% smaller

regional total emissions, in line with the observed negative bias of the TROPOMI version 1.2 product compared to the OMI QA4ECV product. TROPOMI DA can provide city scale emission estimates, which were within 10% differences with other high-resolution analyses for several limited areas, while providing a globally consistent analysis. These results demonstrate that TROPOMI DA improves global analyses of NO₂ and ozone, which would also benefit studies on detailed spatial and temporal variations in ozone and nitrate aerosols and the evaluation of bottom-up NOₓ emission inventories.

## 1 Introduction

Satellite measurements from the Global Ozone Monitoring Experiment (GOME) (Burrows et al., 1999), the Scanning Imaging Absorption Spectrometer for Atmospheric Chartography (SCIAMACHY) (Bovensmann et al., 1999), the Ozone Monitoring Instrument (OMI) (Levelt et al., 2006), and GOME-2 (Callies et al., 2000) have provided long-term global pictures of tropospheric NO₂ columns since 1996. Tropospheric NO₂ is important for air quality, atmospheric chemistry, and climate change

as the main precursor of tropospheric ozone and nitrate aerosols (IPCC, 2021). Although these measurements have provided unprecedented information on global and regional NO₂ variations associated with changes in human and natural activity, their spatial coverage and accuracy limited their ability for a range of applications. Since October 2017, the Tropospheric Monitor-



ing Instrument (TROPOMI) onboard the Sentinel-5 Precursor (Veefkind et al., 2012) has been measuring tropospheric $NO_2$ columns at higher spatial resolutions of $7 \times 3.5$ km$^2$ ($5.5 \times 3.5$ km$^2$ after 6 August 2019) and improved signal-to-noise (S/N)

ratio, compared to previous satellite measurements such as OMI (Eskes et al., 2019; van Geffen et al., 2019).

Satellite $NO_2$ observations have proven useful for constraining $NO_x$ emissions, for instance, through fitting downwind line densities (e.g., Lin et al., 2012; de Foy et al., 2015; Liu et al., 2016) and chemical transport modelling (e.g., Stavrakou et al., 2013; Ding et al., 2015; Miyazaki et al., 2017). Using TROPOMI $NO_2$, surface $NO_x$ emissions have been estimated at high spatial and temporal resolutions, but studies are mostly limited to specific areas at point source to urban scales (Beirle et al.,

2019; Goldberg et al., 2019; Lorente et al., 2019; van der A et al., 2020; Huber et al., 2020; Lange et al., 2021). In recent studies, TROPOMI $NO_2$ retrievals have also been used to provide a detailed understanding of regional and global emission reductions during the COVID-19 lockdowns (Ding et al., 2020; Miyazaki et al., 2020b, 2021; Kim et al., 2021; Zhang et al., 2021). These studies demonstrate the great potential of TROPOMI $NO_2$ for improving the spatial distribution and temporal variability of emissions. Nevertheless, its relative advantage over previous satellite measurements, such as OMI $NO_2$, in emission estimations

for different regions of the world has not been clearly addressed in a quantitative and consistent manner.

Impacts of individual measurements can be evaluated using state-of-the-art data assimilation (DA) techniques, which have widely been used in numerical weather forecast applications (e.g., Gelaro and Zhu, 2009). Chemical DA systems have been used to address measurement impacts on atmospheric composition analysis, including the evaluation of air pollutant emissions (Fortems-Cheiney et al., 2009; Barré et al., 2014, 2015; Emili et al., 2014; Miyazaki et al., 2012b, 2017, 2019; Zhang et al.,

2019). A multi-constituent chemical DA system developed by our group assimilates multiple satellite measurements simultaneously to improve emissions and concentrations of various species (e.g., Miyazaki et al., 2017, 2020a; Sekiya et al., 2021), which allows us to evaluate the relative value of TROPOMI and OMI retrievals in a consistent framework.

In this study, we compared concentration and emission analyses derived from the assimilation of TROPOMI and OMI tropospheric $NO_2$ retrievals, which simultaneously optimize tropospheric $NO_2$, ozone concentrations, and $NO_x$ emissions at

$0.56°$ resolution for the globe. Although this resolution is still insufficient to resolve point source to urban scales, it has the advantage of providing globally–consistent analyses on a megacity scale (Sekiya et al., 2021). The DA analyses were validated against assimilated and independent observations. The systematic comparison of TROPOMI DA and OMI DA reveals relative advantages of DA using TROPOMI over OMI, which benefit studies in particular on the evaluation of bottom-up emission inventories and formation processes of ozone and nitrate aerosols. The remainder of this paper is structured as follows. Section

2 describes the observation data used for the assimilation and validation and the DA system. Section 3 validates tropospheric $NO_2$ concentration analyses against assimilated and independent observations. Sections 4 and 5 present surface $NO_x$ emission analyses and their impacts on the ozone analyses, respectively. Section 6 provides a summary of the study.



## 2 Data and methods

### 2.1 TROPOMI and OMI satellite observations of tropospheric $NO_2$ for assimilation

The TROPOMI and OMI instruments are ultraviolet/visible nadir-scanning solar-backscatter spectrometers (Levelt et al., 2006; Veefkind et al., 2012). The local equator crossing time is approximately 13:40 LT (local time) for both instruments. The TROPOMI and OMI ground pixel sizes are $3.5 \times 7$ km$^2$ ($3.5 \times 5.5$ km$^2$ after 6 August 2019) and $13 \times 24$ km$^2$, respectively. TROPOMI and OMI provide nearly global daily coverage. We used the TROPOMI $NO_2$ unofficial reprocessing product (version 1.2 beta), which is similar to the official version 1.2.2 reprocessing product (van Geffen et al., 2020), and the OMI

QA4ECV version1.1 product (Boersma et al., 2017, 2018) for the period 1 April–31 May 2018. These products were retrieved based on the differential optical absorption spectroscopy (DOAS) approach using the same wavelength window of 405–465 nm, with slight differences in the detailed settings, such as the formulation of modeled reflectance, the fitting methods, and the intensity offset correction (van Geffen et al., 2020). The TROPOMI slant column density (SCD) error for a single pixel is 30% (20% after August 6, 2019) lower than that of the OMI retrievals (van Geffen et al., 2020). A priori $NO_2$ profiles for

TROPOMI and OMI were obtained from the TM5-MP data assimilation system at $1° \times 1°$ resolution. Temporal changes in row anomalies (after 2007), stripes, and instrument radiometric degradation increase the uncertainty of the OMI $NO_2$ SCD by 1–2% per year and decrease the coverage area fraction (Schenkeveld et al., 2017; Zara et al., 2018). Therefore, the relative advantages of TROPOMI over OMI in 2018 need to be evaluated with caution.

The TROPOMI retrievals with quality assurance (QA) values of $> 0.75$ were used, which corresponds to good quality

retrievals over (nearly) cloud free scenes. This screening criteria are similar to the criteria applied for OMI: cloud radiance rfaction (CRF) of $< 0.5$, solar zenith angle (SZA) of $< 81°$, surface albedo of $< 0.3$, quality flag of $= 0$, and ratio of tropospheric air mass factor (AMF) to geometric AMF of $> 0.05$. For OMI, retrievals affected by row anomalies were excluded using a quality flag. Cloud-covered scene retrievals with CRF of $> 0.5$ were separately used for optimizing lightning $NO_x$ sources, following the method proposed by Miyazaki et al. (2014).

Negative biases (by up to 50%) against surface remote sensing observations in the TROPOMI versions 1.2 and 1.3 products were reported by Verhoelst et al. (2021). However, a large fraction of the negative biases might arise from the vertical profile shape of $NO_2$ assumed for retrievals, as reported by Dimitropoulou et al. (2020) for Uccle, Belgium. Compared to the OMI QA4ECV product, the tropospheric $NO_2$ column in the TROPOMI versions 1.2 and 1.3 products are systematically lower especially for winter, as reported by Lambert et al. (2021), which is largely attributed to a negative cloud height bias in the Fast

Retrieval Scheme for Clouds from Oxygen absorption band (FRESCO) implementation (van Geffen et al., 2021).

### 2.2 Independent observations for validation

Vertical profiles and surface concentrations of $NO_2$ and ozone derived from TROPOMI DA and OMI DA were validated against independent observations. The DA analysis fields at the closest time to measurement were sampled using two-hourly analysis output, and then linearly interpolated to the observation locations from the surrounding grids in the horizontal and



vertical directions. Vertical profiles were compared by averaging data within each vertical pressure bin, namely 7 bins from 850 (800–900) to 250 (200–300) hPa.

### 2.2.1 NASA ATom aircraft-campaign observations

Vertical profiles of $NO_2$ were obtained from the NASA Atmospheric Tomography mission 4 (ATom-4) aircraft campaign (Wofsy et al., 2018). The ATom-4 campaign was conducted using a NASA DC-8 aircraft from 24 April to 21 May 2018. The
DC-8 flight tracks covered regions between 85°S and 83°N over the Pacific, Atlantic, and United States. The $NO_2$ concentrations were measured via chemiluminescence with an overall uncertainty of 20 pptv (Weinheimer et al., 1994). To evaluate the DA performance for vertical $NO_2$ profiles over polluted areas, we used data averaged over coastal regions of the western United States (117.25–122.5°W, 32–37°N) from three flights on 24 and 27 April, and on 21 May.

### 2.2.2 Surface in-situ observations

We used surface $NO_2$ and ozone concentrations from 3,255 sites over Europe obtained from the European air quality database (AirBASE) of the European Environmental Agency (EEA), 404 sites over the United States obtained from the Air Quality System (AQS) of the United States Environmental Protection Agency (US EPA), and 1,246 sites over Japan obtained from Japanese continuous measurement data of general air pollution at ground level compiled by the National Institute of Environmental Studies (NIES). We excluded sites in high-traffic and industrial locations, because the 0.56°-resolution model grids
cannot resolve $NO_2$ enhancement at roadside and individual point sources. For AirBASE and AQS, sites with station types of "Industrial" and "Traffic" and with land use of "INDUSTRIAL" and "MOBILE" were excluded, respectively. For Japan, we excluded measurement sites for automobile exhaust gases. More than 97% of the observed $NO_2$ concentrations used in this study were measured by commercial chemiluminescence analyzers, with typical measurement errors of 1–5% (Gluck et al., 2003). These analyzers overestimate the ambient $NO_2$ concentrations, because the measurements contain interference
from reactive nitrogen compounds other than $NO_2$ (e.g., Dickerson et al., 2019). Thus, correction factors proposed by Lamsal et al. (2008) using concentration analyses of $HNO_3$, PAN, and $\sum$ ANs were applied to the observations derived from the commercial chemiluminescence analyzers in the manner described by Sekiya et al. (2021):

$$CF = \frac{NO_2}{NO_2 + \sum ANs + 0.95 \times PAN + 0.35 \times HNO_3},$$

(1)

where $\sum$ AN is the sum of all alkyl nitrate concentrations, and PAN is the peroxyacetyl nitrate concentrations.

### 2.2.3 Ozonesonde observations

The observed vertical profiles of ozone were obtained from the World Ozone and Ultraviolet Data Center (WOUDC, http://www.woudc.org), Southern Hemisphere Additional Ozonesondes (SHADOZ; Sterling et al. (2018); Thompson et al. (2017); Witte et al. (2017, 2018) and the National Oceanic and Atmospheric Administration (NOAA) Earth System Research Laboratory (ESRL) Global Monitoring Division (GMD, ftp://ftp.cmdl.noaa.gov/ozwv/ozone). We used 127 profiles at 19 stations for





the northern extratropics (20–90°N), 45 profiles at 9 stations for the tropics (20°S–20°N), and 36 profiles at 7 stations for the southern extratropics (20–90°S).

## 2.3 Data assimilation system

### 2.3.1 CHASER chemical transport model

We used the global chemical transport model, CHASER V4.0, at a resolution of 0.56° with 32 vertical layers (Sudo et al., 2002; Sekiya et al., 2018) as the forecast model, which simulates tracer transport, emission, dry and wet deposition, and chemical processes (92 species and 262 reactions) including the ozone-$HO_x$-$NO_x$-CO-VOCs system. The meteorological fields simulated by the dynamical and physical modules of CHASER (i.e., MIROC-AGCM atmospheric general circulation model (K-1 model developers, 2004)) were nudged to the 6-hourly ERA-Interim reanalysis data (Dee et al., 2011) with a relaxation time of 5 days for temperature and 0.7 days for horizontal winds, and used in the chemical module of CHASER at every time step (1–4 min). We demonstrated that increasing model resolution from the conventional resolution (2.8°) to 0.56° resolution substantially improves the model performance over polluted regions (Sekiya et al., 2018).

The a priori surface $NO_x$ emissions were obtained from the HTAPv2.2 anthropogenic emission inventory (at 0.1° resolution) for 2010 (Janssens-Maenhout et al., 2015), the Global Fire Emission Database (GFED) version 4.1s monthly-based biomass burning emission inventory (at 0.25° resolution) for 2018 (Randerson et al., 2018), and the Global Emission Initiative (GEIA) soil $NO_x$ emission inventory (at 0.5° resolution) (Yienger and Levy, 1995). The a priori lightning $NO_x$ sources were calculated in the model at each model time step using the parameterization proposed by Price and Rind (1992).

### 2.3.2 Ensemble Kalman filter data assimilation

We developed a state-of-the-art chemical DA system (e.g., Miyazaki et al., 2019, 2020a) using the local ensemble transform Kalman filter (LETKF) technique (Hunt et al., 2007). The LETKF uses an ensemble model forecast to estimate background error covariance assuming that the background ensemble perturbations sample the forecast model errors. The background ensemble model fields were converted into observation space by applying the observation operator which includes a spatial interpolation operator, and an averaging kernel. The inclusion of averaging kernels in the observation operator describes the vertically-dependent sensitivities and removes the influence of a-priori profile shape (Eskes and Boersma, 2003). The analysis ensemble mean $\overline{x^{\mathrm{a}}}$ was obtained by combining the background ensemble mean $\overline{x^{\mathrm{b}}}$ and assimilated observations $y^{\mathrm{o}}$ with relative weights, which were determined using background and observation error covariance matrices $\mathbf{X}^{\mathrm{b}}$ and $\boldsymbol{R}$, respectively:

$$\overline{x^{\mathrm{a}}} = \overline{x^{\mathrm{b}}} + \mathbf{X}^{\mathrm{b}} \tilde{\boldsymbol{P}}^{\mathrm{a}} \left(\boldsymbol{Y}^{\mathrm{b}}\right)^{T} \boldsymbol{R}^{-1} \left(y^{\mathrm{o}} - \overline{y^{\mathrm{b}}}\right), \qquad (2)$$

where $\tilde{\boldsymbol{P}}^{\mathrm{a}}$ is the local analysis error covariance in the ensemble space, while $\overline{y^{\mathrm{b}}}$ and $\boldsymbol{Y}^{\mathrm{b}}$ are the background ensemble mean and error covariance in the observation space, respectively. The local analysis error covariance was estimated as

$$\tilde{\boldsymbol{P}}^{\mathrm{a}} = \left[\frac{(k-1)\boldsymbol{I}}{1+\Delta} + \left(\boldsymbol{Y}^{\mathrm{b}}\right)^{T} \boldsymbol{R}^{-1} \boldsymbol{Y}^{\mathrm{b}}\right]^{-1}, \qquad (3)$$



where $\Delta$ is a covariance inflation factor ($= 7\%$ per DA cycle) and $k$ is the ensemble size (32 or 64 in this study, see Table 1 for details).

Surface and lightning $NO_x$ emissions were estimated based on a state argumentation method (e.g., Evensen, 2009) using the relationship between $NO_2$ concentrations and $NO_x$ emissions in the background error covariance matrix generated based on ensemble model simulations. The initial a priori error was set as 40% and 60% for the surface and lightning $NO_x$ sources, 

respectively. In the analysis step, the standard deviation of emission ensembles was artificially inflated to a predefined minimum value obtained through sensitivity calculations (i.e., 56% of a priori emissions) to prevent covariance underestimation.

Our previous study (Sekiya et al., 2021) demonstrated that DA improvements were larger by factors of 1.5–3 at 0.56° resolution than at 2.8° resolution over polluted regions in comparison to the assimilated $NO_2$ observations. This high resolution leads to reduced spatial representativeness errors (due to an increased average coverage fraction per grid cell at 0.56° resolution 

by a factor of two, compared to 2.8° resolution). The 0.56°-resolution ensemble model simulation also generates background error covariance matrix which describes small(0.56°)-scale features. Because of distinct non-linear transport and chemical processes, assimilation considering the background error covariance would also be essential for making the best use of observational information. The multi-constituent DA system have been used to assimilate ozone, $NO_2$, CO, $SO_2$, and $HNO_3$ (Miyazaki et al., 2020a). Nevertheless, in this study, only TROPOMI and OMI $NO_2$ were assimilated to emphasize the impact 

of assimilation of tropospheric $NO_2$ retrievals.

### 2.3.3 Super-observation approach

The super-observation approach (Eskes et al., 2003; Miyazaki et al., 2012a) was used for generating satellite observation data representative to the model grid size (i.e., 0.56°). The super-observation approach can minimize spatial representativeness errors for spatially varying concentrations of short-lived tracers on sub-grid scales, such as $NO_2$, compared to the thinning 

approach which randomly selects an observation per grid cell (Boersma et al., 2016). The resolution of super-observation was set to be identical to the forecast model resolution. The super-observation concentration was generated by averaging all data within a super-observation grid cell, while applying a weighting function based on the coverage area of overlap with the super-observation grid cell. The super-observation error was calculated as a combination of measurement and spatial representativeness errors $\sqrt{\sigma_m + \sigma_r}$. In our approach, the super-observation measurement error $\sigma_m$ was estimated as

$$\sigma_m = \sqrt{(1-c)\sum_{i=1}^{n} w_i^2 \sigma_{m,i}^2 + c \sum_{i=1}^{n} (w_i \sigma_{m,i})^2}, \tag{4}$$

where $\sigma_{m,i}$ is the mean measurement error at individual pixels related to total slant column density (SCD), stratosphere–troposphere separation (STS), and tropospheric AMF, $c$ is the error correlation coefficient among the individual retrieval data for these error components, and $n$ is the number of measurements with non-zero overlap with the chosen grid cell. This approach explicitly accounts for spatial correlations, $c$, between observation errors which depends on the error sources, compared to 

the conventional approach used in Miyazaki et al. (2012a). It is supposed that observation errors related to total SCD and tropospheric AMF contain larger random components (by 85%, i.e., $c = 0.15$) than those related to STS (by 0%, i.e., $c = 1.0$).





The spatial representativeness error $\sigma_r$ was calculated as a function of coverage area fraction in the same way as Boersma et al. (2016).

### 2.4 Experimental setup

As summarized in Table 1, four DA runs from April–May (61 days) were performed. Firstly, we conducted two DA calculations for TROPOMI and OMI separately at an ensemble size of 64. This comparison was used to investigate how TROPOMI DA improves agreements with assimilated and independent observations, compared to OMI DA (Sections 3.2–3.4, 4, and 5). Secondly, we compared OMI DA calculations for two different years (2005 and 2018) at an ensemble size of 32. This demonstrates the impacts of OMI instrumental degradation and row anomalies , which significantly reduce daily coverage (c.f.,

Section 2.1), on the DA performance (Section 3.5). In addition, a control model simulation without any DA was conducted to measure the DA impacts in each case. We chose the calculation period of April–May 2018 because of the AToM-4 aircraft-campaign data availability (see Section 2.2.1). Furthermore, systematic biases between the TROPOMI and OMI retrievals are known to be smaller in the summer season than those in the winter season (Lambert et al., 2020). We analyzed the DA results for the period 15 April–31 May after a 2 week-long spin-up.

## 3 Validation results

### 3.1 Data characteristics

Super-observation concentrations and errors can affect DA results, which are compared in Figure 1 and Table 2. The TROPOMI and OMI super-observation concentrations were well correlated ($r = 0.82$ over 60°S–60°N) during April–May 2018, with lower concentrations in TROPOMI by 15% averaged over 60°S–60°N without applying averaging kernels of each other. The

mean super-observation errors and mean relative super-observation errors (i.e., errors divided by concentrations) in TROPOMI averaged over 60°S–60°N were compared to those in OMI. The mean super-observation errors were 33% smaller in TROPOMI than in OMI, while the mean relative super-observation errors were 16% smaller in TROPOMI. These differences mainly result from improvements in SCD-related errors associated with improved S/N ratio of TROPOMI data, reduced random error components by increasing spatial resolution of TROPOMI data (i.e., an increasing number of observations per super-

observation grid cell; see equation 4), and smaller TROPOMI stripes. Over polluted regions, because individual retrieval uncertainties scale with tropospheric column amounts, the lower mean concentrations in TROPOMI than in OMI also led to the smaller super-observation errors in TROPOMI (by 33%). As an exception, over remote regions, reduced S/N ratio in SCD, rather than the lower concentrations, explain the smaller super-observation errors (by 32%) in TROPOMI. Over some remote areas, such as northern high latitudes, the Tarim basin, the tropical Pacific Ocean, and southern midlatitudes, relative

errors were larger in TROPOMI than in OMI (Figure 1i). The larger TROPOMI relative errros over these areas are influenced by dominant contribution of the uncertainties in stratospheric column for TROPOMI because of reduced random error components in TROPOMI and the assumption of spatial correlation $c = 1$.





The spatial coverage per super-observation grid cell of TROPOMI (72%) was larger than those of OMI (69%) mainly because of OMI row anomalies, which led to smaller spatial representativeness errors of TROPOMI (7%) than those of OMI (10%). The mean relative super-observations errors of OMI were 8% smaller in 2005 than in 2018 (figure not shown), which is attributed to the temporal changes in OMI row anomalies, stripes, and instrument radiometric degradation (see Section 2.1). The averaging kernel values in the lower troposphere (below 850 hPa) were higher by 44% in TROPOMI averaged over 60°S–60°N than those in OMI, because mean CRF over 60°S–60°N is 15% smaller in TROPOMI due to better resolving small-scale cloud-free scenes.

## 3.2 Self-consistency

The performance of TROPOMI DA and OMI DA was confirmed by the $\chi^2$ test (Ménard and Chang, 2000; Zupanski and Zupanski, 2006). $\chi^2$ value is diagnosed from the ratio of the Observation-minus-Forecast (OmF; i.e., $\mathbf{y}^o - H(\mathbf{x}^b)$) to estimated error covariance in the observational space ($\mathbf{H}\mathbf{P}^b\mathbf{H}^T + \mathbf{R}$) as

$$\mathbf{Y} = \frac{1}{\sqrt{N}}(\mathbf{H}\mathbf{P}^b\mathbf{H}^T + \mathbf{R})^{-1/2}(\mathbf{y}^o - H(\mathbf{x}^b)) \tag{5}$$

$$\chi^2 = \text{trace } \mathbf{Y}\mathbf{Y}^T. \tag{6}$$

The mean values of estimated $\chi^2$ over polluted regions ($> 1 \times 10^{15}$ molecules cm$^{-2}$) after inflation factor tuning was 0.99 for TROPOMI DA, which is close to the ideal value of 1. The mean $\chi^2$ of 1.17 for OMI suggests underestimated background error covariance or observation errors.

We also evaluated the self-consistency with the assimilated observations based on reductions in root-mean-square error (RMSE) by DA ($\Delta$RMSE) using daily maps sampled at observation locations as

$$\Delta\text{RMSE} = -\left(\sqrt{\frac{\sum_{i=1}^{N}(A(c_{assim}) - V)^2}{N}} - \sqrt{\frac{\sum_{i=1}^{N}(A(c_{ctl}) - V)^2}{N}}\right) \bigg/ \left(\sqrt{\frac{\sum_{i=1}^{N}(A(c_{ctl}) - V)^2}{N}}\right) \tag{7}$$

where $V$ and $A$ are the observed tropospheric NO$_2$ column and corresponding averaging kernels, respectively, used for DA; $c_{assim}$ and $c_{ctl}$ are NO$_2$ concentration fields obtained from the DA runs and control model simulations, respectively; and $N$ is the number of super-observation data. The level of significance of $\Delta$RMSE was determined using the Mann-Whitney U test (Mann and Whitney, 1947).

As shown in Figure 2 and Table 3, the RMSE for TROPOMI DA over 60°S–60°N was reduced by 54% compared to that for the control model simulation, with larger RMSE reductions over polluted regions (by 60%) than over remote regions (by 37%). The RMSE reductions were substantial over most regions in the tropics and northern midlatitudes, whereas improvements are not clear over the northern high latitudes, Tarim Basin, Arabian Sea, northern Australia, South America, and parts of the southern mid-latitudes. Mean RMSE reductions were larger for TROPOMI DA than OMI DA (by 38%). The differences in RMSE reductions between TROPOMI DA and OMI DA over the tropics and northern midlatitudes were statistically significant at the 95% confidence level. These differences can be explained by the reduced relative super-observation errors in TROPOMI. In contrast, the differences in RMSE reduction between TROPOMI DA and OMI DA were statistically insignificant over most regions with larger relative super-observation errors in TROPOMI.





The two-dimensional histogram of grid-level relative super-observation errors and RMSE reductions (Figures 3a and 3b) shows clear decreases in RMSE reductions with increasing relative super-observation errors for both TROPOMI and OMI DA. Steep RMSE decreases occurred around relative super-observation errors of 20–50%, which reflected areas over and downwind of polluted regions. Over polluted regions, observational information is more effectively assimilated into the model, because of the large uncertainty (i.e., background error covariance) of estimated $NO_x$ emissions over these regions. As shown in Figure 3c,

mean relative super-observation errors at individual grids were smaller than those in TROPOMI in OMI by 16%. Corresponding to these smaller super-observation errors, the mean RMSE reductions by TROPOMI DA at individual grids (by 54%) were larger than those by OMI DA (by 38%), with large differences in frequency of RMSE reductions between TROPOMI DA and OMI DA for RMSE reductions of > 10% (Figure 3d). These results confirm that improved RMSE reductions by TROPOMI DA compared to OMI DA can be attributed to the reduced relative super-observation errors in TROPOMI. Meanwhile, the obtained

result suggests that DA efficiency by TROPOMI is determined by the amount and quality of TROPOMI data, regardless of the TROPOMI low bias.

### 3.3    Validation against independent observations

#### 3.3.1    ATom aircraft-campaign data

Figure 4 and Table 4 compare the vertical profiles of $NO_2$ with the ATom-4 aircraft campaign observations on 24 and 27

April, and 21 May when the DC-8 aircraft flew over coastal areas of the western United States. The control model simulation overestimated the observed concentrations in the lower troposphere (700–900 hPa) by factors of 1.4–4 in all cases, while underestimating the $NO_2$ concentrations in the middle and upper troposphere (300–700 hPa) by 48–70%, except on 27 April. The positive model biases were particularly large at 750 hPa on 24 April and at 850 hPa on 27 April and 21 May. The use of 2010 anthropogenic $NO_x$ emissions could explain the positive model biases. In addition, on 24 April, the simulated planetary

boundary layer (PBL) height was 30% higher than that in the ERA-Interim reanalysis, which could in turn increase $NO_2$ bias at 750 hPa.

On 24 April, TROPOMI DA increased negative bias at 850 hPa, while it reduced positive bias at 750 hPa, which could also be attributable to model biases in PBL height. The mean bias in the lower troposphere (below 700 hPa) was largely reduced by TROPOMI DA (by 84%) on 24 April. The improvements were small (by 17%) on 27 April when the DC-8 aircraft $NO_2$

measurements were conducted in the early morning before the TROPOMI overpass time, whereas TROPOMI DA reduced positive model biases by 78% in the lower troposphere on 21 May. In the middle and upper troposphere, TROPOMI DA reduced the model biases by 12–53%. These bias reductions were larger by 52–70% for the lower troposphere and by up to 31% for the middle and upper troposphere in TROPOMI DA than OMI DA, except for the lower troposphere on 27 April. Because of the large variability in the observed concentrations, these differences in bias were statistically insignificant based

on a two-sample t test, except for the upper troposphere on 21 May.





### 3.3.2 Surface in-situ data

Surface in-situ observation data at 14:00 LT was used for validation to evaluate assimilation impacts just after their overpass times. Validation was conducted after filtering out model grids where water bodies cover >50% of a grid box area using annual Moderate Resolution Imaging Spectrodadiometer (MODIS) land cover data (Friedl and Sulla-Menashe, 2015) for 2018, considering large representativeness errors. Over Europe, the regional mean model bias and RMSE of $NO_2$ were −18% and 145%, respectively (absolute errors are shown in Table 5). The model biases vary with regions, with positive biases of 12–115% over the United Kingdom (UK), Belgium, and the Netherlands and negative biases of 42–78% over Italy, Serbia, and Romania (Figure 5). Over the United States, regional mean model bias and RMSE were 37% and 268%, respectively, with larger positive biases over urban areas such as New York, Los Angeles, and Chicago. The regional mean bias and RMSE over Japan were −23% and 124%, respectively.

TROPOMI DA reduced the regional RMSE over Europe by 29%, with larger RMSE reductions by 45% and 47% over the UK and the Netherlands, respectively, reflecting improvements in spatial and temporal variability by TROPOMI DA (Figure 6). Because of the small RMSEs in the control model simulation, RMSE reductions by TROPOMI DA were not obvious over Italy, Spain, and Portugal. Over the United States, TROPOMI DA reduced the regional mean bias and RMSE by 46% and 50%, respectively. In contrast to the large RMSE reductions over the eastern United States and western coastal areas, RMSEs increased over Colorado and Wyoming again due to the small RMSEs in the control model simulations. Over Japan, TROPOMI DA reduced RMSE by 23%, but increased negative model bias by 68%. Error reductions were smaller in OMI DA overall. The RMSE over Europe was increased by OMI DA by 5% mainly due to the increased errors over the Netherlands. Over the United States and Japan, the RMSE reductions for megacities such as New York, Los Angeles, and Tokyo were 25–70% larger in TROPOMI DA than in OMI DA. The regional RMSE reduction was comparable between the two runs (by 47% for the United States and 20% for Japan by OMI DA).

### 3.4 Regional performance over Los Angeles

The magnitude of improvements by DA can be affected by meteorological conditions (e.g., Miyazaki et al., 2019). We evaluated impacts of meteorological conditions on the self-consistency over Los Angeles where both independent surface and aircraft-campaign observations were available (Sections 3.3.1 and 3.3.2). During 15 April–31 May, southwesterly winds were predominant over Los Angeles, while wind speed varied (c.f., Figure 8). As shown in Figure 7, the RMSE over Los Angeles city (black rectangles of Figure 7) were reduced by TROPOMI DA in windy conditions (wind speed > 2.5 m s$^{-1}$) by 46% and calm conditions by 37%. Over Los Angeles city, the RMSE reductions by TROPOMI DA were larger under the windy conditions (by a factor of 1.3) compared with OMI DA, with statistical significance at the 99% confidence level; the RMSE reductions were comparable under the calm conditions (within 5%). The TROPOMI measurements with high vertical sensitivity (i.e., averaging kernels) in the lower troposphere captured the dilution of $NO_2$ over Los Angeles city during the windy conditions better than OMI, which resulted in better improvements by TROPOMI DA than OMI DA under the windy conditions. The vertical sensitivity of TROPOMI in the lower troposphere over Los Angeles city was 36% higher than that of OMI,



reflecting a smaller CRF in TROPOMI than in OMI (by 17%), because TROPOMI has higher spatial resolution and better
resolves small-scale cloud-free scenes over Los Angeles. In contrast, the super-observation errors and quantities in TROPOMI
and OMI during the windy conditions were comparable to those during the calm conditions (figure not shown). Meanwhile,
over the surrounding areas of Los Angeles city, RMSE reductions by TROPOMI DA and OMI DA were comparable under
both the windy and calm conditions.

As summarized in Table 6, positive model bias in $NO_2$ concentrations against in-situ observations for 15 April–31 May
2018 was 33%. Temporal correlation coefficient between observed and simulated concentrations was 0.49. TROPOMI DA
introduced negative bias, whereas improving the temporal correlation to 0.63. RMSEs were reduced by 37% and 26% during
the calm and windy conditions, respectively. The negative bias was larger in TROPOMI DA than in OMI DA, whereas temporal
correlation coefficients in TROPOMI DA ($r = 0.63$) were larger than those in OMI DA ($r = 0.25$). The RMSE reductions
by TROPOMI DA were 8% larger than those by OMI DA under the windy conditions, whereas the RMSE reductions by
TROPOMI DA and OMI DA were comparable under the calm conditions. These results suggest similar improvements by
TROPOMI DA compared to OMI DA under the windy and calm conditions, while meteorological conditions slightly affect
the magnitude of improvements in $NO_2$ concentrations by TROPOMI DA compared to OMI DA.

### 3.5 Impact of OMI instrumental degradation

Temporal changes in OMI row anomalies, stripes, and instrument radiometric degradation from 2005 to 2018 could affect OMI
DA results. Thus, we compared OMI DA results between 2005 and 2018. As summarized in Table 3, the RMSE reduction over
polluted regions in 2018 (by 48%) was larger than that in 2005 (by 41%) with statistical significance at the 99% confidence
level. The multi-year difference in DA performance is likely driven by inter-annual variations in meteorological conditions
rather than by OMI degradation. Over Europe, the United States, and China, the number of cloud-free scenes in 2018 was
increased by 11–19% compared to those in 2005. In contrast, RMSE reductions over remote regions were similar (23% in
2018 and 25% in 2005). Such interannual changes in cloud cover can affect the overall OMI DA performance, which needs
to be considered in the TROPOMI and OMI comparison results for 2018. Nevertheless, the improvements against assimilated
observations by TROPOMI DA (by 54%) were larger than those by OMI DA for both years 2005 (by 34%) and 2018 (by 38%).
TROPOMI DA clearly shows better performance compared to OMI DA for the periods before instrumental degradation, even
when considering inter-annual variations in meteorological conditions.

### 4 $NO_x$ emission estimates

The top-down estimates provided by TROPOMI DA significantly differed from the a priori emissions (Figure 9 and Table 7).
TROPOMI DA tends to decrease emissions over the eastern United States, China, northern India, and Central Africa. Large
positive increments (by 42% on average) were found over regions where soil emissions are dominant ($> 50\%$ in a priori
emissions), such as over remote areas of Spain, Turkey, the Midwest United States, Kazakhstan, and the Sahel regions. This
suggests underestimated soil emissions in a prior inventories, as commonly reported by previous studies (Vinken et al., 2014;





Oikawa et al., 2015; Visser et al., 2019). The country and regional total emissions were decreased by 14% in the United States, 38% China, 17% in India, and 22% in Central Africa, and increased by 12% in Europe, 39% in the Middle East, and 44 % in Southeast Asia.

The global total $NO_x$ emissions were 15% smaller in TROPOMI DA than in OMI DA, with 3–18% smaller regional total

emissions for polluted regions and 22–23% smaller regional total emissions for biomass burning regions, reflecting the low bias of TROPOMI retrievals compared to OMI retrievals. The low bias of the TROPOMI retrievals compared to the OMI retrievals also affects OH concentrations. Assimilation of lower $NO_2$ retrievals, through $NO_x$ emission and $NO_2$ concentration optimization, led to weaker chemical production of $HO_x$ and conversion from $HO_2$ to OH. This effect resulted in 2–21% smaller regional-mean OH concentrations in the lower troposphere in TROPOMI DA, except for South Africa. In contrast,

differences in the regional total emissions over India and the Middle East between TROPOMI DA and OMI DA were small (4–5%), reflecting small differences in regional-mean concentrations between the TROPOMI and OMI retrievals (4–6% lower in TROPOMI). Compared to the EDGARv5 (Crippa et al., 2019) and REASv3.2 (Kurokawa and Ohara, 2020) bottom-up emission inventories for 2015, the regional total emissions from TROPOMI DA and OMI DA over major polluted regions, except for Europe, were smaller by 17–35% and 9–21%, respectively. These results suggest that the emission estimates from

OMI DA are closer to the EDGARv5 and REASv3.2 bottom-up emission inventories than those from TROPOMI DA (using the TROPOMI v1.2beta product).

The $NO_x$ emissions derived from TROPOMI DA were compared with previous estimates over large urban areas based on statistical fits of $NO_2$ line density data with the exponentially modified Gaussian (EMG) function using TROPOMI $NO_2$ (Beirle et al., 2019; Lorente et al., 2019; Goldberg et al., 2019; Lange et al., 2021). We focused on large urban areas where at least two

estimates were available. For this comparison, a posterior emissions from our TROPOMI DA estimates were integrated inside a square of $100{\times}100$ $km^2$ centered on the selected urban area, while the uncertainty information was obtained from the analysis ensemble spreads. As summarized in Table 8, our estimates were in good agreement with the previous estimates within 10% differences for Riyadh, Chicago, and New York compared to the estimates from Beirle et al. (2019) and Lange et al. (2021). Nevertheless, lower emissions in our estimates by 18–66% over Chicago, New York, and Toronto than the estimates from

Goldberg et al. (2019) could be explained by the difference in the TROPOMI $NO_2$ AMF calculation, which was replaced by Environment and Climate Change Canada (ECCC) with their high-resolution regional CTM and the MODIS surface reflectance (Griffin et al., 2019). For Paris, our estimate in late spring was lower by 35% and 41% than those by Lorente et al. (2019) and Lange et al. (2021), respectively, but analyzed for different time periods in winter and annually, respectively. Increases in $NO_x$ emissions during cold seasons are because of residential heating (Lorente et al., 2019).

Cloud-covered scenes of satellite $NO_2$ retrievals were used to optimize lightning $NO_x$ sources following the method of Miyazaki et al. (2014), which provide important constraints on tropospheric chemistry including ozone (e.g., Boersma et al., 2005; Miyazaki et al., 2014; Allen et al., 2021). Because of its high spatial resolution, TROPOMI $NO_2$ retrievals offer the potential for better resolving small-scale cloud-covered scenes (Marais et al., 2021) and constraining lightning $NO_x$ sources. As a result, the difference between TROPOMI DA and OMI DA can be attributed to 75–92% higher vertical sensitivities above

the cloud height for cloud-covered scenes (CRF $>$ 0.5). As shown in Table 7, the global total production of lightning NO





estimated by TROPOMI DA was 13% larger than that estimated by OMI DA, with larger regional total production by 14–52% over North and South America, Southeast Asia, the Atlantic, and Indian Ocean. The impacts of TROPOMI on lightning $NO_x$ source estimation will be investigated in more detail in a separate study.

## 5  Impacts on ozone analysis

The $NO_2$ DA plays an important role in improving the representation of tropospheric chemistry, including ozone. We evaluated the relative values of TROPOMI and OMI $NO_2$ DA on surface and tropospheric ozone analysis.

### 5.1  Validation using surface in-situ data

Daily maximum 8-h average (MDA8) ozone concentrations were validated using surface in-situ observation data in the same manner as $NO_2$. As summarized in Table 5, the regional mean bias and RMSE of the control model simulation against surface
in-situ ozone observations over Europe were 22% and 29%, respectively, with large RMSEs over southern Europe (Figure 10). Over the United States, the mean model bias and RMSE were 10% and 22%, respectively, reflecting large model biases over the eastern United States. The mean bias and RMSE over Japan were 7% and 18%, respectively. Positive model biases in surface ozone over polluted regions are commonly reported in other global CTMs (Schnell et al., 2015; Turnock et al., 2020).

TROPOMI DA increased the mean bias and RMSE of surface ozone over Europe by 14% and 8%, respectively, with
large error increases over southern Europe. Increased bias and RMSE were also found over the western United States. Many factors can lead to increased errors in ozone, including model errors in ozone precursors' emissions other than $NO_x$, chemical processes, and meteorological processes such as PBL vertical mixing. For example, ozone responses to $NO_x$ emissions strongly depend on the choice of CTMs (Miyazaki et al., 2020c), which affects the impacts of $NO_x$ emission corrections on ozone analyses. Over the eastern United States, RMSEs were typically reduced by 5–10 ppbv by TROPOMI DA, resulting in improved
bias by 14% and RMSE by 16% at country scale. Over Japan, the mean bias and RMSE were reduced by TROPOMI DA by 54% and 7%, respectively.

For most regions, better agreement with surface ozone data was obtained from TROPOMI DA than from OMI DA. OMI DA resulted in larger increases in the positive bias and RMSE over Europe by 32% and 22%, respectively. Over the United States, the mean bias and RMSE were slightly increased by OMI DA by 8% and 1%, respectively. The bias reduction over
Japan by TROPOMI DA was larger than that by OMI DA (by 18%), while RMSE over Japan was increased by OMI DA (by 4%). The better agreement in surface ozone by TROPOMI DA coincides with that in surface $NO_2$ (c.f., Section 3.3.2). This confirms that the better representation of $NO_2$ through assimilation of advanced $NO_2$ satellites is essential to improve surface ozone analysis for many regions of the world. Meanwhile, any biases in satellite $NO_2$ retrievals affect surface ozone analysis. Surface ozone analysis bias could be improved by using updated retrievals with reduced TROPOMI $NO_2$ negative biases.



## 5.2 Validation using ozonesonde data

Here we focus on the $NO_2$ DA impacts on free tropospheric ozone. Mean negative model biases of ozone at 500 and 800 hPa against ozonesonde observations were 9.5% and 3.8%, respectively, over the 20–90°N band, 15.7% and 3.6% over the tropics (20°S–20°N), and 14.4% and 20.6% over the 20–90°S band. The RMSEs at 500 and 800 hPa were 16% and 23%, respectively, over the 20–90°N band, 26% and 31% over the tropics, and 23% and 18% over the 20–90°S band (Table 9).

TROPOMI DA greatly reduced the mean model biases and RMSE by 98% and 24% at 500 hPa, respectively, and 82% and 14% at 800 hPa over the 20–90°N band. Over the 20–90°S band, there were reductions of 67% and 40% at 500 hPa, respectively, and 70% and 35 % at 800 hPa. In contrast, it introduced large positive biases in the tropics. The reductions in model bias and RMSEs over the 20–90°N and S bands provided by TROPOMI DA were larger than those by OMI DA (by 24–91% and 12–22%, respectively). The increases in model biases and RMSEs over the tropics by TROPOMI DA were smaller than those by OMI DA. The larger differences at 500 hPa over the tropics than over the extratropics can be attributed to smaller $NO_x$ emission estimates over biomass burning regions in TROPOMI DA than those in OMI DA, through upward transport of ozone and $NO_x$ and chemical processes. In addition, the tropospheric ozone burden over 60°N–60°S was smaller in TROPOMI DA (291 $TgO_3$) than in OMI DA (304 $TgO_3$), while estimates from both TROPOMI DA and OMI DA were within the 287–311 $TgO_3$ range of satellite-based estimates (i.e., the OMI/MLS, OMI-SAO, OMI-RAL, IASI-FORLI, and IASI-SOFRID satellite products) for the period 2014–2016 (Gaudel et al., 2018).

## 6 Summary and conclusion

We compared DA analyses of $NO_2$, ozone concentrations, and $NO_x$ emissions derived from the assimilation of the TROPOMI and OMI tropospheric $NO_2$ column retrievals. To generate observation data representative to the model grid size, we employed a super-observation approach that explicitly accounts for spatial correlations between observation errors. Because of 16% smaller relative super-observation errors in TROPOMI than in OMI, the DA self-consistency, as measured by RMSE reductions against the assimilated observations, was improved in TROPOMI DA by 54%, which was larger than OMI DA (by 38%). Agreements against the independent ATom-4 aircraft-campaign and surface in-situ $NO_2$ data were also improved by 12–84% and 23–50%, respectively, which was larger than those for OMI DA (by up to 70% and 47%, respectively) for many cases. The improved $NO_2$ led to improved agreement with surface in-situ MDA8 ozone over United States and Japan in TROPOMI DA (by 7–40%) than in OMI DA (by 1–22%). Agreements with ozonesonde data at 500 and 800 hPa were also improved by TROPOMI DA by 14–40% for most regions, except for the tropics, which was larger than those by OMI DA (by 12–22%).

Global total $NO_x$ emission was increased from 43.5 Tg N $yr^{-1}$ in a priori emissions to 46.2 Tg N $yr^{-1}$ by TROPOMI DA, which was 15% smaller than those derived from OMI DA (54.2 TgN), with 3–23% smaller regional total emissions for major polluted and biomass burning areas. The city-scale emissions derived from TROPOMI DA were generally consistent with previous estimates using limited-area high resolution analyses (within 10% differences for Riyadh, New York, and Chicago). The global emission estimates constrained by the more accurate and dense TROPOMI measurements provide complementary information about emission variability, especially where accurate and detailed information on activity data and emission



factors is missing when developing bottom-up inventories (Elguindi et al., 2020). This would also benefit model simulations of tropospheric ozone (e.g., Miyazaki et al., 2019; Visser et al., 2019; Bae et al., 2020; Qu et al., 2020), and estimations of nitrate aerosols and their deposition flux (Nowlan et al., 2014; Geddes and Martin, 2017). These improvements are important for productivities and diversities of terrestrial and marine ecosystems.

The DA performance comparisons provide a systematic evaluation of TROPOMI and OMI retrievals, independent from their averaging kernels and a priori profiles. The improved agreements with independent observations in TROPOMI DA demonstrate the importance of improved spatial coverage and reduced retrieval uncertainty for many science applications. Meanwhile, validation against surface $NO_2$ measurements showed lower bias in TROPOMI retrievals compared to OMI retrievals by 15% for the United States, Europe, and Japan. The smaller estimated $NO_x$ emissions also confirm the low biases in TROPOMI $NO_2$ relative to OMI $NO_2$ globally, which also affected ozone analysis. This systematic bias is largely attributed to a negative cloud height bias in the FRESCO cloud retrieval algorithm (van Geffen et al., 2021). New versions of the TROPOMI $NO_2$ product were introduced in December 2020 (version 1.4) and July 2021 (version 2.2). As shown by van Geffen et al. (2021), these new versions largely remove the bias with the OMI QA4ECV product. Meanwhile relative super-observation errors of the new version (2.2.0) are comparable to those in version 1.2.2 (Appendix A). These differences in the new version products would not affect the main conclusions of this study and improve $NO_x$ emission and ozone analyses.

The evaluation of individual satellite measurement through DA integration provides unique and detailed information on possible errors, including their spatio-temporal structures, which in turn supports satellite retrieval developments. Meanwhile, application of bias correction in DA analysis is essential for the combined use of observational information from multiple sensors, including those from other polar orbit satellites such as OMPS and advanced geostationary satellites such as GEMS, TEMPO, and Sentinel-4.

*Code availability.* The source codes are not publicly available because of license restriction. The source code is available from Kengo Sudo (kengo@nagoya-u.jp) upon request. The source code for the data assimilation system is available from Kazuyuki Miyazaki (Kazuyuki.Miyazaki@jpl.nasa.gov) upon request.

*Data availability.* The data assimilation results are available online (https://figshare.com/projects/Sekiya_et_al_2021/126770). The TROPOMI and OMI satellite retrievals are publicly available at the TEMIS website (http://www.temis.nl). The AirBASE data, AQS data, and Japanese continuous measurement data were provided by the EEA (https://www.eea.europa.eu/data-and-maps/data/aqereporting-8), the US EPA (https://www.epa.gov/aqs), and NIES (https://www.nies.go.jp), respectively. The ATom-4 observation data were taken from the NASA Ames Earth Science Project Office (https://espo.nasa.gov/atom). Ozonesonde observation data were obtained from WOUDC (https://woudc.org), SHADOZ (https://tropo.gsfc.nasa.gov/shadoz/), and NOAA ESRL GMD ( ftp://ftp.cmdl.noaa.gov/ozwv/ozone). MCD12C1 MODIS/Terra+Aqua Land Cover Type data (Yearly L3 Global 0.05Deg CMG) were obtained from NASA Earthdata web site (https://earthdata.nasa.gov).





*Author contributions.* TS, KM, and HE designed this study. TS conducted data assimilation calculations and analyzed the results. HE provided the TROPOMI and OMI $NO_2$ data. All the coauthors commented upon and helped improve the manuscript.

465     *Competing interests.* The authors declare that they have no conflict of interest.

*Acknowledgements.* This work was supported by MEXT (JPMXP1020351142) as "Program for Promoting Researches on the Supercomputer Fugaku" (Large Ensemble Atmospheric and Environmental Prediction for Disaster Prevention and Mitigation), and by JSPS KAKENHI grants (18H01285 and 18KK0102). A part of the research was conducted at the Jet Propulsion Laboratory, California Institute of Technology, under a contract with NASA. We acknowledge the use of tropospheric $NO_2$ column data from TROPOMI and OMI sensors in the
470    ESA Sentinel-5 Precursor and the NASA Aura satellite missions, respectively. Earth Simulator was used for simulation and data assimilation calculations with the support of JAMSTEC.



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



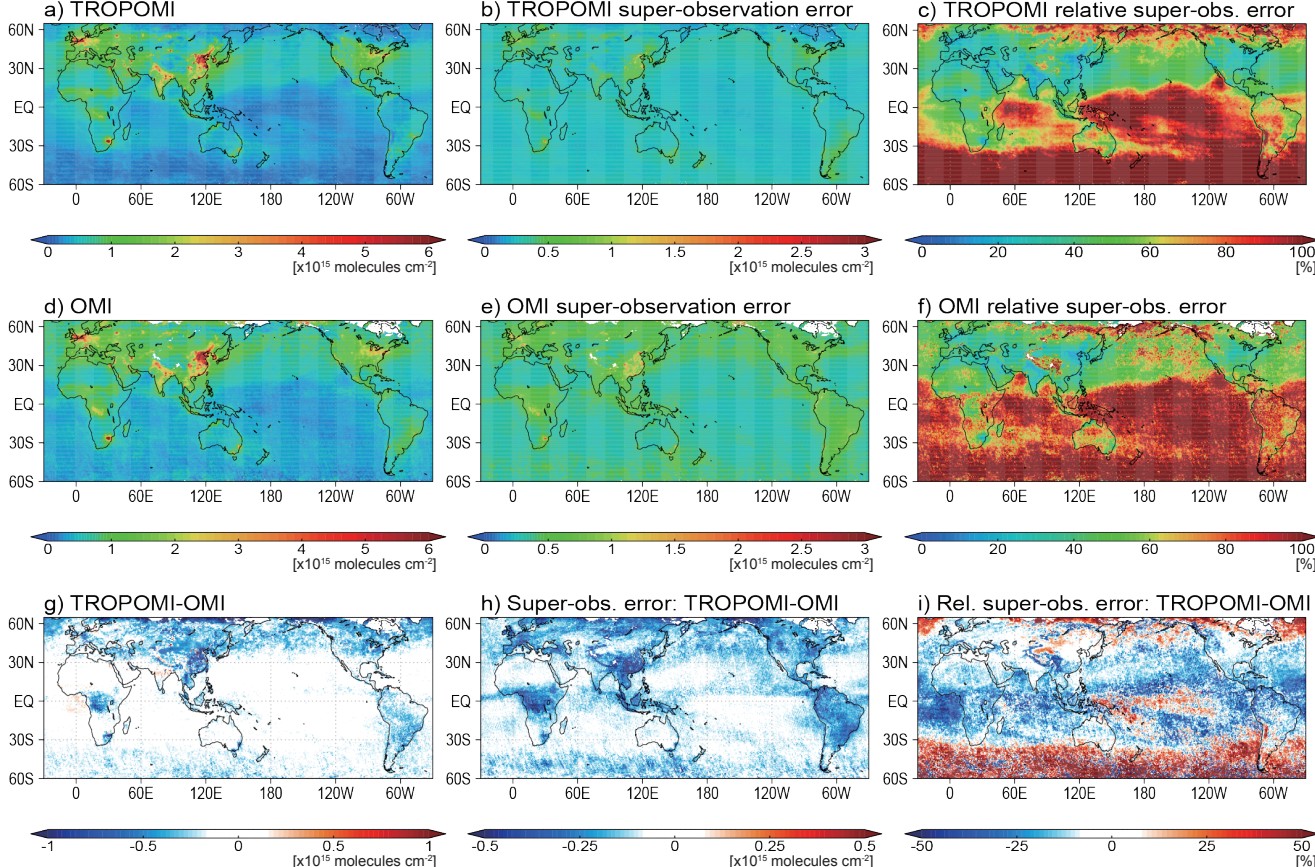

**Figure 1.** Global distribution of tropospheric $NO_2$ column (left), super-observation errors (middle), and relative super-observation errors (right) obtained from the Tropospheric Monitoring Instrument (TROPOMI; top) and the Ozone Monitoring Instrument (OMI; middle) from April–May 2018, and the differences between TROPOMI and OMI (bottom). The units of the tropospheric $NO_2$ column, super-observation errors, and relative super-observation errors are $\times 10^{15}$ molecules cm$^{-2}$, $\times 10^{15}$ molecules cm$^{-2}$, and percentage (%), respectively.





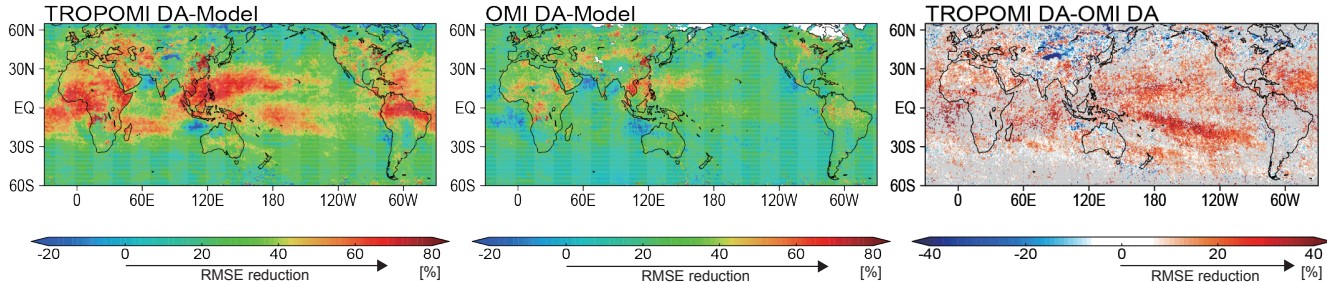

**Figure 2.** Root-mean-square error (RMSE) reduction for tropospheric $NO_2$ concentration fields against assimilated observations by data assimilation (DA) (%) obtained from Tropospheric Monitoring Instrument (TROPOMI) DA (left) and Ozone Monitoring Instrument (OMI) DA (middle), and the differences between them (right). For the right panel, grids with a gray color indicate differences between RMSE reductions by TROPOMI DA and OMI DA that are statistically insignificant at the 95% confidence level using the Mann-Whitney U test.

**Table 1.** List of data assimilation and control model simulation runs performed in this study.

| No. | Experiment | Period | Assimilated observation | Ensemble size |
|-----|------------|--------|-------------------------|---------------|
| 1 | TROPOMI DA (2018, N=64) | April–May, 2018 | TROPOMI v1.2beta | 64 |
| 2 | OMI DA (2018, N=64) | April–May, 2018 | OMI QA4ECV v1.1 | 64 |
| 3 | OMI DA (2018, N=32) | April–May, 2018 | OMI QA4ECV v1.1 | 32 |
| 4 | OMI DA (2005) | April–May, 2005 | OMI QA4ECV v1.1 | 32 |
| 5 | Control model simulation | April–May, 2018<br>April–May, 2005 | | |

**Table 2.** Mean tropospheric $NO_2$ column ($\times\ 10^{-15}$ molecules cm$^{-2}$), super-observation error ($\times\ 10^{-15}$ molecules cm$^{-2}$), relative super-observation error (%) over 60°S–60°N in the Tropospheric Monitoring Instrument (TROPOMI) from April to May 2018, and the Ozone Monitoring Instrument (OMI) from April to May in 2005 and 2018. Values in brackets are calculated from TROPOMI and OMI data with co-location criteria of < 60 km in space and < 2 h in time.

| Satellite observations | Tropospheric column | Super-observation error | Relative super-observation error |
|------------------------|---------------------|-------------------------|----------------------------------|
| TROPOMI | 0.52 (0.54) | 0.28 (0.27) | 53 (51) |
| OMI (2018) | 0.63 (0.64) | 0.40 (0.38) | 63 (60) |
| OMI (2005) | 0.57 | 0.33 | 58 |

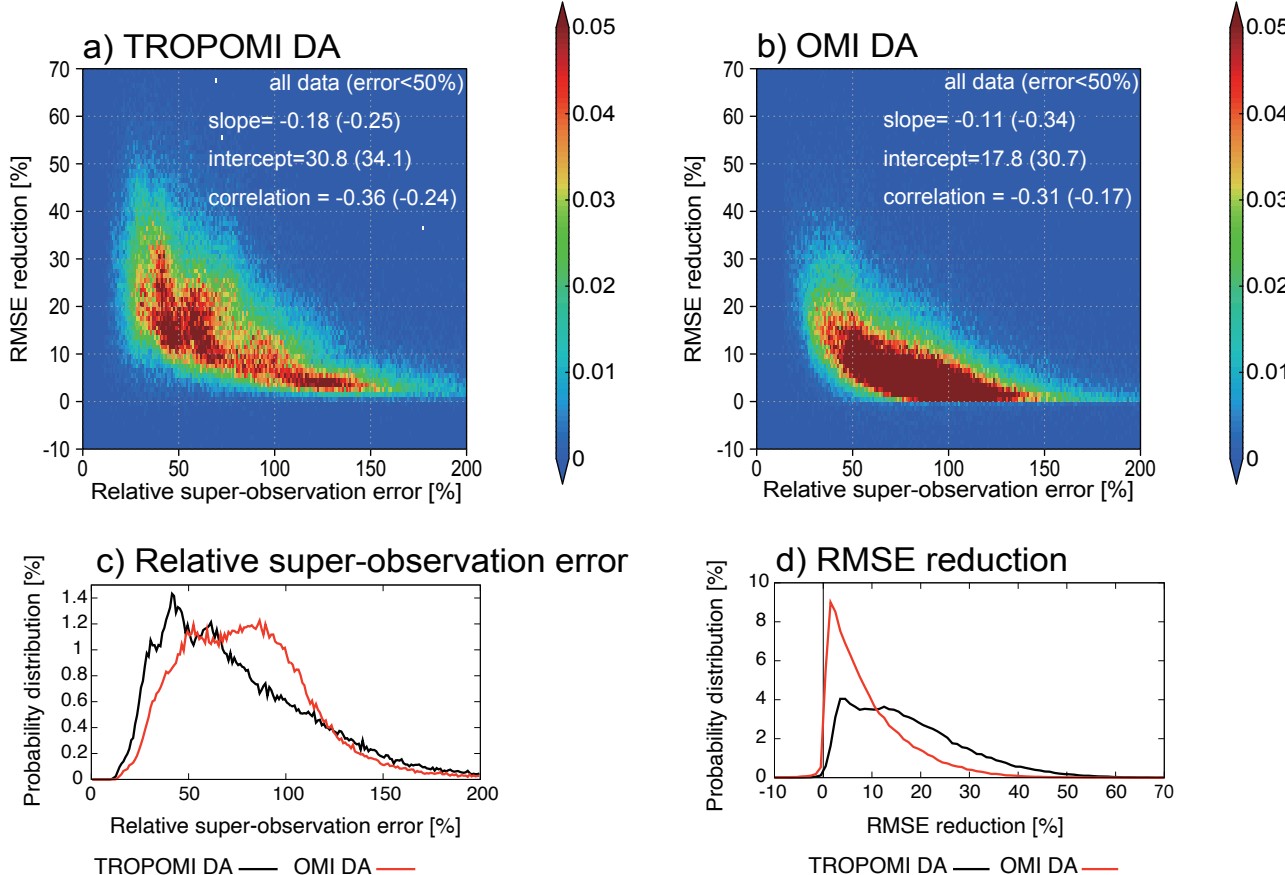

**Figure 3.** Two-dimensional (2-D) histogram (%) as a function of relative super-observation errors (%) and root-mean-square error (RMSE) reductions (%) for tropospheric $NO_2$ column made to data assimilation (DA) (top row) obtained from Tropospheric Monitoring Instrument (TROPOMI) DA (left) and Ozone monitoring Instrument (OMI) DA (right). The values in the top panels are regression coefficients, intercepts, and correlation coefficients between relative super-observation errors and RMSE reductions against assimilated observations by DA on the grid scale. The values in brackets were calculated using data with relative super-observation errors of $< 50\%$. One-dimensional (1-D) histogram as a function of relative super-observation errors (left) and RMSE reductions (right) are exhibited in the bottom row. The black and red lines are taken from TROPOMI DA and OMI DA, respectively.



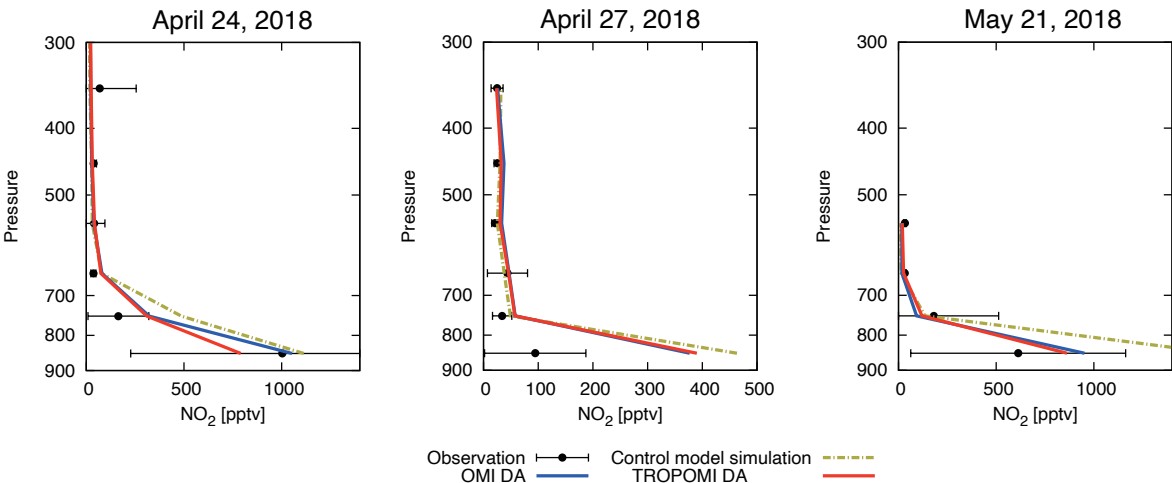

**Figure 4.** Vertical profiles of $NO_2$ (pptv) on 24 April (left), 27 April (middle), and 21 May (right) over coastal areas of the western United States (117.25–122.5°W, 32–37°N). The results were obtained from the ATom-4 aircraft campaign observations (black), Tropospheric Monitoring Instrument (TROPOMI) data assimilation (DA) (red), Ozone Monitoring Instrument (OMI) DA (blue), and the control model simulation (yellow). The error ranges are the standard deviations of individual values in each pressure bin.

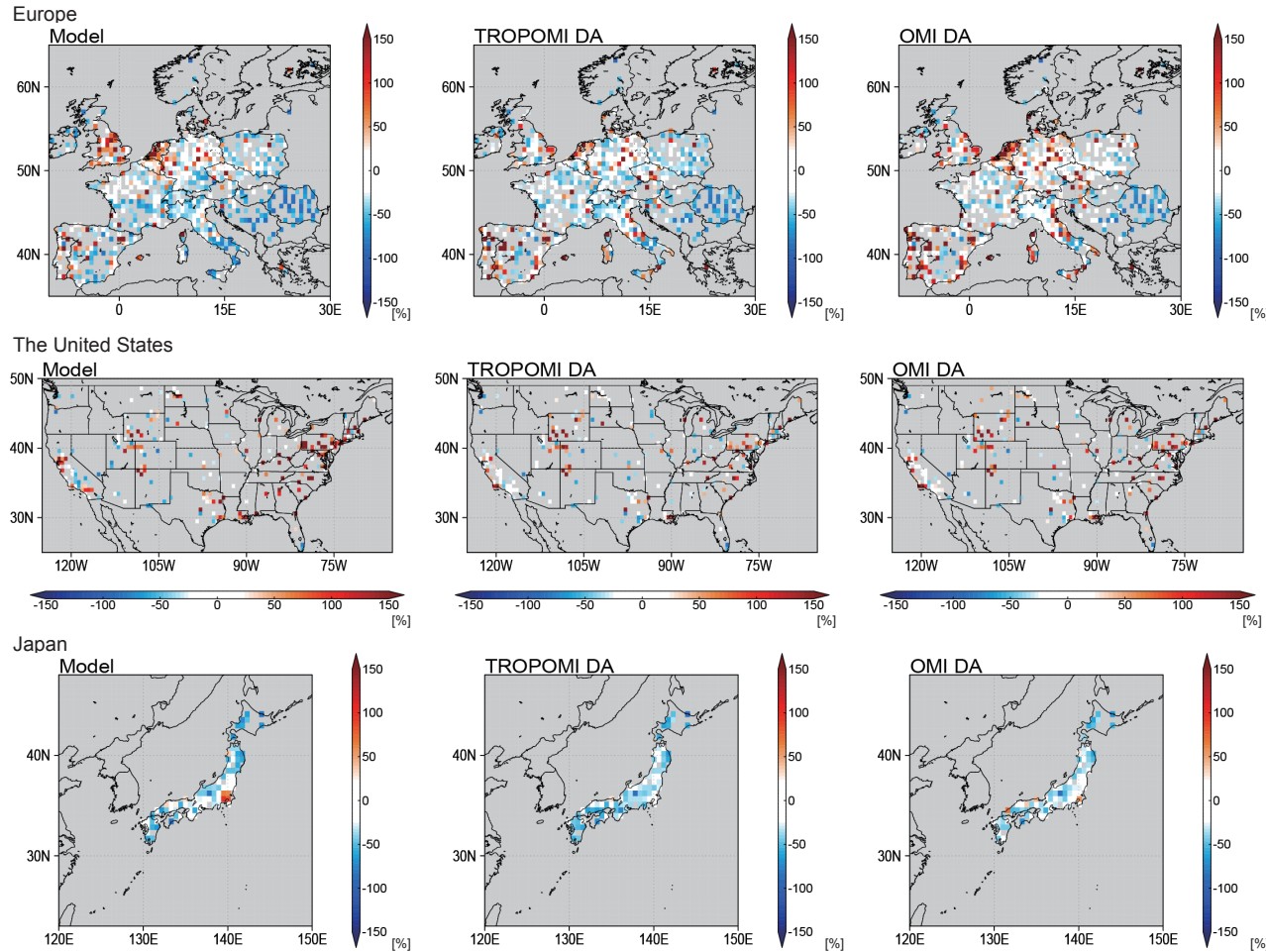

**Figure 5.** Mean model biases against surface in-situ observations for surface NO₂ concentrations (%) at 14:00LT (local time) derived from the control model simulation (left column), Tropospheric Monitoring Instrument (TROPOMI) data assimilation (DA) (middle column) and Ozone Monitoring Instrument (OMI) DA (right column) over Europe (top), the United States (middle), and Japan (bottom). The values are mapped onto 0.56° resolution grids.



**Figure 6.** Root-mean-square errors (RMSEs) against surface in-situ observations for surface $NO_2$ concentration fields (ppbv) at 14:00 LT (local time) in the control model simulation (left column) and their reductions (%) by Tropospheric Monitoring Instrument (TROPOMI) data assimilation (DA) and Ozone Monitoring Instrument (OMI) DA (middle and right columns, respectively) over Europe (top), the United States (middle), and Japan (bottom). The values are mapped onto 0.56° resolution grids. For the middle and right columns, grids with open circles indicate RMSE reductions that are statistically significant at the 95% confidence level using the Mann-Whitney U-test.



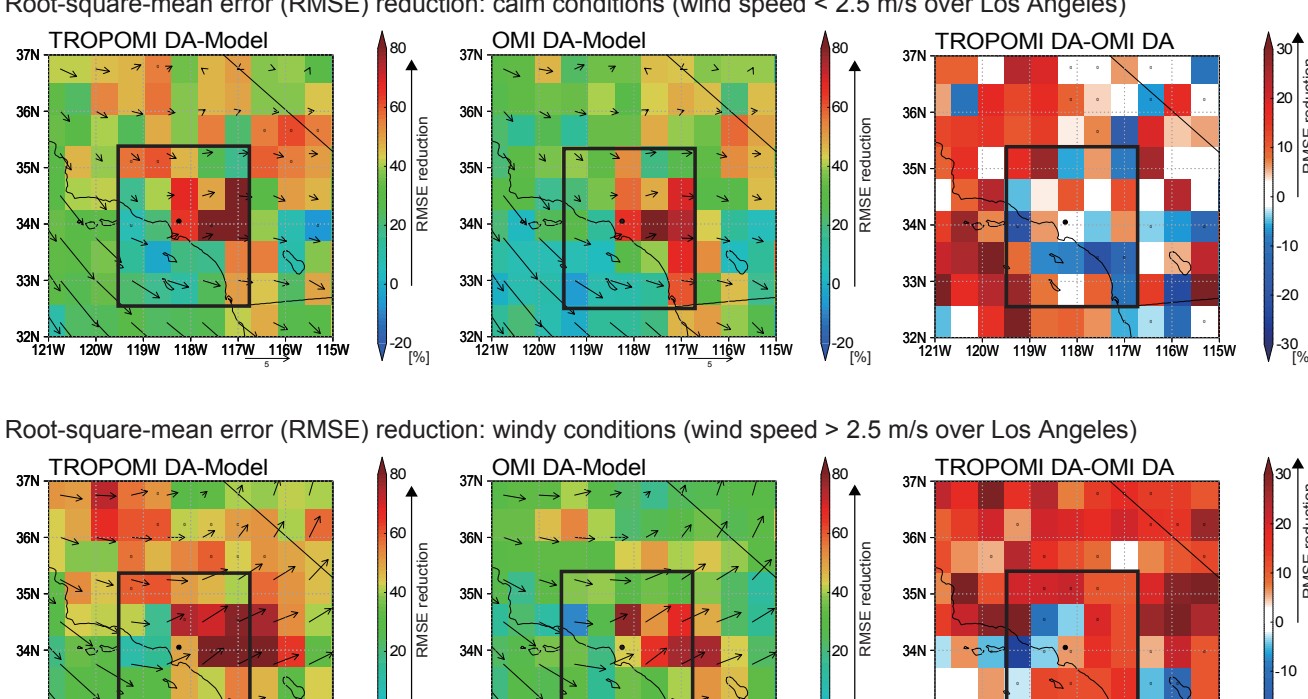

**Figure 7.** Root-mean-square error (RMSE) reduction for tropospheric $NO_2$ concentration fields against assimilated observations by data assimilation (DA) over Los Angeles under windy (wind speed > 2.5 m s$^{-1}$, top row) and calm (bottom row) conditions. The left, middle, and right columns show Tropospheric Monitoring Instrument (TROPOMI) DA, Ozone Monitoring Instrument (OMI) DA, and the differences between them, respectively. The unit is percentage (%). Arrows in the left and middle columns show surface winds derived from ERA-Interim reanalysis data. For the right column, grids with open circles indicate the differences in RMSE reductions between TROPOMI DA and OMI DA that are statistically significant at the 95% confidence level using the Mann-Whitney U-test. The black circle indicate the location of Los Angles city center.





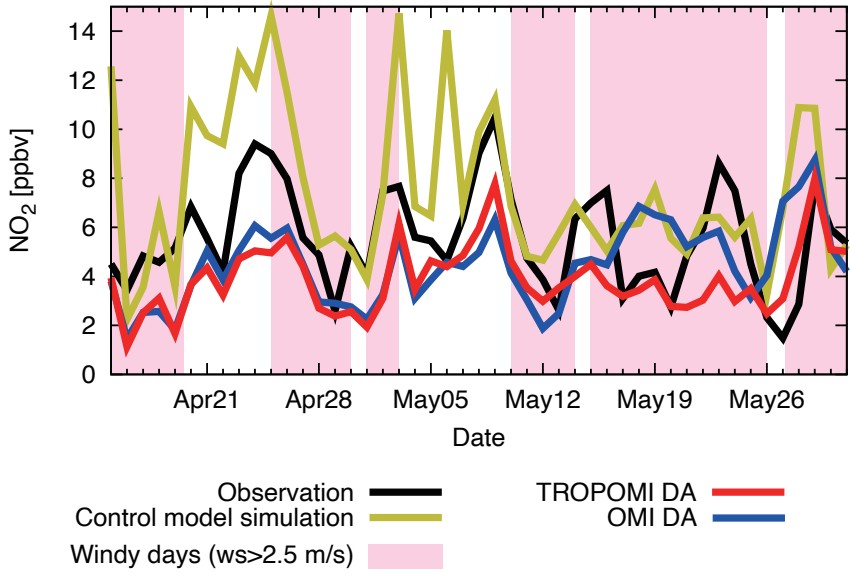

**Figure 8.** Surface $NO_2$ concentrations (ppbv) at 14:00LT (local time) over Los Angeles. The results were obtained from in-situ observations (black), Tropospheric Monitoring Instrument (TROPOMI) data assimilation (DA) (red), Ozone Monitoring Instrument (OMI) DA (blue), and the control model simulation (yellow). The periods filled in pink are windy conditions (wind speed $> 2.5$ m s$^{-1}$).

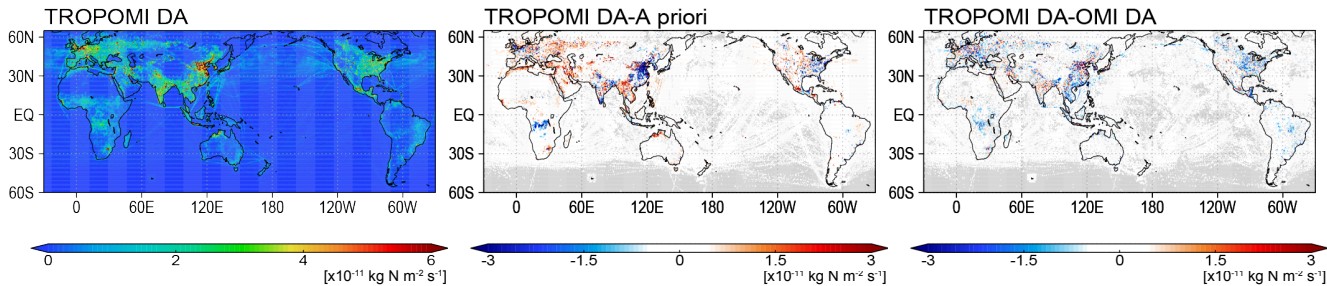

**Figure 9.** Global distributions of top-down $NO_x$ emission estimates ($\times 10^{-11}$ kg N m$^{-2}$ s$^{-1}$) provided by Tropospheric Monitoring Instrument (TROPOMI) data assimilation (DA) from 15 April–31 May 2018 (left), and the differences between TROPOMI DA and a priori emissions (middle) and between TROPOMI DA and Ozone Monitoring Instrument (OMI) DA (right). For the middle and right panels, grids with a gray color indicate the differences that are statistically insignificant at the 95% confidence level using the Mann-Whitney U-test.





**Figure 10.** Root-mean-square errors (RMSEs) against surface in-situ observations (ppbv) for daily maximum 8-h average (MDA8) ozone concentrations in the control model simulation (left), and their reductions by Tropospheric Monitoring Instrument (TROPOMI) data assimilation (DA) and Ozone Monitoring Instrument (OMI) DA (middle and right, respectively) over Europe (top), the United States (middle), and Japan (bottom). The values are mapped onto 0.56° resolution grids. For the middle and right columns, grids with open circles indicate the RMSE reductions that are statistically significant at the 95% confidence level using the Mann-Whitney U-test.

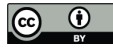



**Table 3.** Root-mean-square error (RMSE) for tropospheric $NO_2$ concentration fields against assimilated observations after data assimilation (DA) (the control model simulation in brackets) and differences in RMSEs between DA and the control model simulation ($\Delta$RMSE) over 60°S–60°N from 15 April–31 May. N denotes the ensemble size. RMSE and $\Delta$RMSE are expressed as $\times 10^{15}$ molecules cm$^{-2}$ and percentage (%), respectively. Polluted regions are defined as the areas where the Tropospheric Monitoring Instrument (TROPOMI) tropospheric $NO_2$ column contains $> 1 \times 10^{15}$ molecules cm$^{-2}$. The ranges of RMSE and $\Delta$RMSE are the standard deviation of the time series.

| Assimilation runs | 60°S–60°N | | Polluted | | Remote | |
| --- | --- | --- | --- | --- | --- | --- |
| | RMSE | $\Delta$RMSE | RMSE | $\Delta$RMSE | RMSE | $\Delta$RMSE |
| TROPOMI DA (2018, N=64) | 0.25±0.03 (0.55±0.06) | -54±12 | 0.65±0.12 (1.61±0.24) | -60±17 | 0.17±0.01 (0.27±0.01) | -37±5 |
| OMI DA (2018, N=64) | 0.40±0.02 (0.65±0.07) | -38±11 | 0.82±0.10 (1.64±0.25) | -50±16 | 0.33±0.01 (0.43±0.02) | -23±5 |
| OMI DA (2018, N=32) | 0.41±0.02 (0.65±0.07) | -37±11 | 0.85±0.11 (1.64±0.25) | -48±17 | 0.33±0.01 (0.43±0.02) | -23±5 |
| OMI DA (2005, N=32) | 0.41±0.04 (0.62±0.07) | -34±12 | 1.00±0.12 (1.69±0.21) | -41±14 | 0.27±0.02 (0.36±0.02) | -25±8 |





**Table 4.** Mean bias (MB) against the ATom-4 aircraft campaign observations in the lower (700–900 hPa) and middle–upper troposphere (300–700 hPa) over coastal areas of the western United States (117.25–122.5°W, 32–37°N) on 24 and 27 April, and 21 May 2018. MB is expressed in pptv. Boldface denotes the best agreement with in-situ observations. The ranges of MB are the standard deviations of individual values in each pressure bin. "Model" indicate the control model simulation.

| | April 24 | | April 27 | | May 21 | |
|---|---|---|---|---|---|---|
| | 700–900 hPa | 300–700 hPa | 700–900 hPa | 300–700 hPa | 700–900 hPa | 300–700 hPa |
| TROPOMI DA | **-33.5±516.0** | **-21.2±144.1** | 166.0±295.1 | **2.8±18.8** | **139.5±412.7** | **-10.6±7.6** |
| OMI DA | 103.5±429.9 | -22.7±144.8 | **159.2±278.8** | 5.3±19.1 | 187.0±471.9 | -15.5±7.3 |
| Model | 216.0±487.4 | -26.4±146.0 | 200.4±419.1 | 3.2±19.3 | 643.3±834.7 | -22.6±7.0 |

**Table 5.** Mean bias (MB) and root-mean-square error (RMSE) for surface $NO_2$ at 14:00LT (local time) and daily maximum 8-h average (MDA8) ozone against in-situ observations during 15 April–31 May 2018. The units of MB and RMSE are ppbv. Boldface denotes the best agreement with in-situ observations. The ranges of MB and RMSE are the standard deviation of the time series. "Model" indicate the control model simulation.

| Region | Run | $NO_2$ | | MDA8 ozone | |
|---|---|---|---|---|---|
| | | MB | RMSE | MB | RMSE |
| Europe | TROPOMI DA | -0.43±0.44 | **1.71±0.54** | 11.15±2.80 | 13.92±2.53 |
| | OMI DA | **-0.12±1.02** | 2.53±0.46 | 12.87±3.37 | 15.73±3.33 |
| | Model | -0.30±0.80 | 2.41±0.46 | **9.75±2.67** | **12.92±2.39** |
| The United States | TROPOMI DA | -0.29±0.51 | **1.93±0.56** | **4.66±3.01** | **9.53±2.90** |
| | OMI DA | **-0.08±0.55** | 2.02±0.57 | 5.90±2.71 | 11.50±4.30 |
| | Model | 0.53±1.55 | 3.87±0.47 | 5.42±3.19 | 11.39±3.71 |
| Japan | TROPOMI DA | -0.80±0.93 | **1.91±0.61** | **1.67±5.24** | **9.63±2.95** |
| | OMI DA | -0.49±0.95 | 1.98±0.61 | 3.00±5.51 | 10.81±3.82 |
| | Model | **-0.47±1.35** | 2.47±0.50 | 3.70±5.30 | 10.36±3.17 |





**Table 6.** Mean bias (MB), temporal correlation coefficient (T-Corr.), and root-mean-square error (RMSE) for surface $NO_2$ at 14:00LT (local time) against in-situ observations from the Air Quality System (AQS) over Los Angeles from 15 April–31 May 2018) under all, calm (wind speed $\leq 2.5$ m s$^{-1}$), and windy conditions. Both MB and RMSE are expressed in ppbv. Boldface denotes the best agreement with in-situ observations. The range of the MB is the standard deviation of the time series. "Model" indicate the control model simulation.

| | All conditions | | | Calm condition | | | Windy condition | | |
|---|---|---|---|---|---|---|---|---|---|
| | MB | T-Corr. | RMSE | MB | T-Corr. | RMSE | MB | T-Corr. | RMSE |
| TROPOMI DA | -1.69±1.60 | **0.63** | **2.33** | -2.08±1.54 | **0.74** | **2.59** | -1.49±1.60 | **0.50** | **2.19** |
| OMI DA | **-1.11±2.28** | 0.25 | 2.54 | **-1.80±2.24** | 0.23 | 2.87 | **-0.76±2.22** | 0.22 | 2.35 |
| Model | 1.88±2.80 | 0.49 | 3.37 | 2.72±3.04 | 0.34 | 4.08 | 1.45±2.57 | 0.45 | 2.95 |

**Table 7.** Global and regional total surface and lightning $NO_x$ emissions (Tg yr$^{-1}$) from 15 April–31 May 2018, taken from a priori emissions, Tropospheric Monitoring Instrument (TROPOMI) data assimilation (DA), Ozone Monitoring Instrument (OMI) DA, EDGARv5 (for 2015)+GFED4 (for 2018) inventories, and REASv3.2 (for 2015)+GFED4 (for 2018) inventories. The ranges are the standard deviations of the time series. A priori lightning emissions are calculated in the control model simulation.

| | A priori | TROPOMI DA | OMI DA | EDGARv5+GFED4 | REASv3.2+GFED4 |
|---|---|---|---|---|---|
| Global | 43.5±0.5 | 46.2±0.9 | 54.2±1.1 | 46.9±1.3 | |
| Europe | 4.1±0.03 | 4.6±0.3 | 5.3±0.1 | 3.9±0.3 | |
| The United States | 5.0±0.03 | 4.3±0.5 | 5.0±0.4 | 5.2±0.2 | |
| China | 7.9±0.4 | 4.9±0.1 | 6.0±0.1 | 7.6±0.1 | 6.6±0.06 |
| India | 3.5±0.004 | 2.9±0.1 | 3.1±0.1 | 3.8±0.03 | 3.6±0.007 |
| Middle East | 2.3±0.004 | 3.2±0.05 | 3.3±0.1 | 2.9±0.04 | |
| South Africa | 0.36±0.003 | 0.34±0.03 | 0.38±0.04 | 0.42±0.001 | |
| Central Africa | 1.8±0.2 | 1.4±0.4 | 1.8±0.4 | 1.4±0.6 | |
| Southeast Asia | 0.9±0.1 | 1.3±0.2 | 1.7±0.2 | 1.2±0.1 | 1.0±0.1 |
| Global lightning | 6.1±0.3 | 6.9±0.1 | 6.1±0.9 | | |





**Table 8.** NO$_x$ emission estimates in large urban areas obtained from this study and previous studies. The unit is Mg/hr. Emissions are averaged for the period 15 April–31 May 2018 for this study, March 2018–November 2020 for Lange et al. (2021), December 2017–October 2018 for Beirle et al. (2019), May–September 2018 for Goldberg et al. (2019), February–June 2018 for Lorente et al. (2019). The ranges of the emissions are emission analysis spreads in this study, while the ranges are the errors estimated by individual previous studies.

| City | This study | Lange et al. (2021) | Beirle et al. (2019) | Goldberg et al. (2019) | Lorente et al. (2019) |
|---|---|---|---|---|---|
| Riyadh (24.6°N, 46.7°E) | 21.5±0.9 | 21.8±0.8 | 23.8 | | |
| Chicago (41.8°N, 87.8°W) | 11.0±2.4 | 12.1±1.1 | | 18.8±5 | |
| New York (40.7°N, 74.0°W) | 14.6±2.5 | 14.7±1.5 | | 17.9±5 | |
| Toronto (43.7°N, 79.4°W) | 4.9±2.9 | 7.6±0.5 | | 14.3±5 | |
| Paris (48.9°N, 2.3°E) | 5.2±2.7 | 8.0±0.5 | | | 8.8 |

**Table 9.** Mean bias (MB) and root-mean-square error (RMSE) for ozone concentrations at 500 and 800 hPa against ozonesonde observations over three latitude bands from 15 April–31 May 2018. The units of MB and RMSE are ppbv. Boldface denotes the best agreement with in-situ observations. "Model" indicate the control model simulation.

| Latitude bands | Run | 500 hPa | | 800 hPa | |
|---|---|---|---|---|---|
| | | MB | RMSE | MB | RMSE |
| | TROPOMI DA | **-0.10** | **12.22** | **0.39** | **8.06** |
| 20–90°N | OMI DA | 0.56 | 12.47 | 1.62 | 8.27 |
| | Model | -6.59 | 16.03 | -2.12 | 9.39 |
| | TROPOMI DA | 7.00 | 10.70 | 5.00 | 9.66 |
| 20°S–20°N | OMI DA | 10.22 | 13.78 | 5.87 | 10.93 |
| | Model | **-5.93** | **10.05** | **-0.93** | **8.01** |
| | TROPOMI DA | **1.51** | **3.88** | **-2.11** | **4.57** |
| 20–90°S | OMI DA | 2.46 | 5.21 | -2.57 | 5.44 |
| | Model | -5.11 | 6.48 | -6.34 | 7.02 |



## Appendix A: Characteristics of the TROPOMI version 1.2 and 2.2 products

Figure A1 compares global distributions of tropospheric $NO_2$ column, super-observation errors, and relative super-observation
errors (i.e., errors divided by concentrations) obtained from the TROPOMI version 1.2 product during September 2018 and the
version 2.2 product during September 2021. We found large differences in tropospheric column amounts between 2018 and
2021 due to algorithm updates and inter-annual changes, while relative super-observation errors in the version 2.2 product over
polluted regions are comparable to those in the version 1.2 product, because individual retrieval uncertainties scale with column
amounts. Over remote regions, large differences in relative super-observation errors at a grid scale are caused by inter-annual
750    variations in the column amounts, because individual retrieval uncertainties do not scale with column amounts due to reduced
S/N ratio over remote regions and stratospheric column-related uncertainties, which are a dominant factor. The differences
in the mean relative super-observation error over 60°N–60°S between the version 2.2 and 1.2 products is within 1.3%, even
though data from different years are compared. These differences are much smaller than the differences between the TROPOMI
version 1.2beta and OMI QA4ECV products.

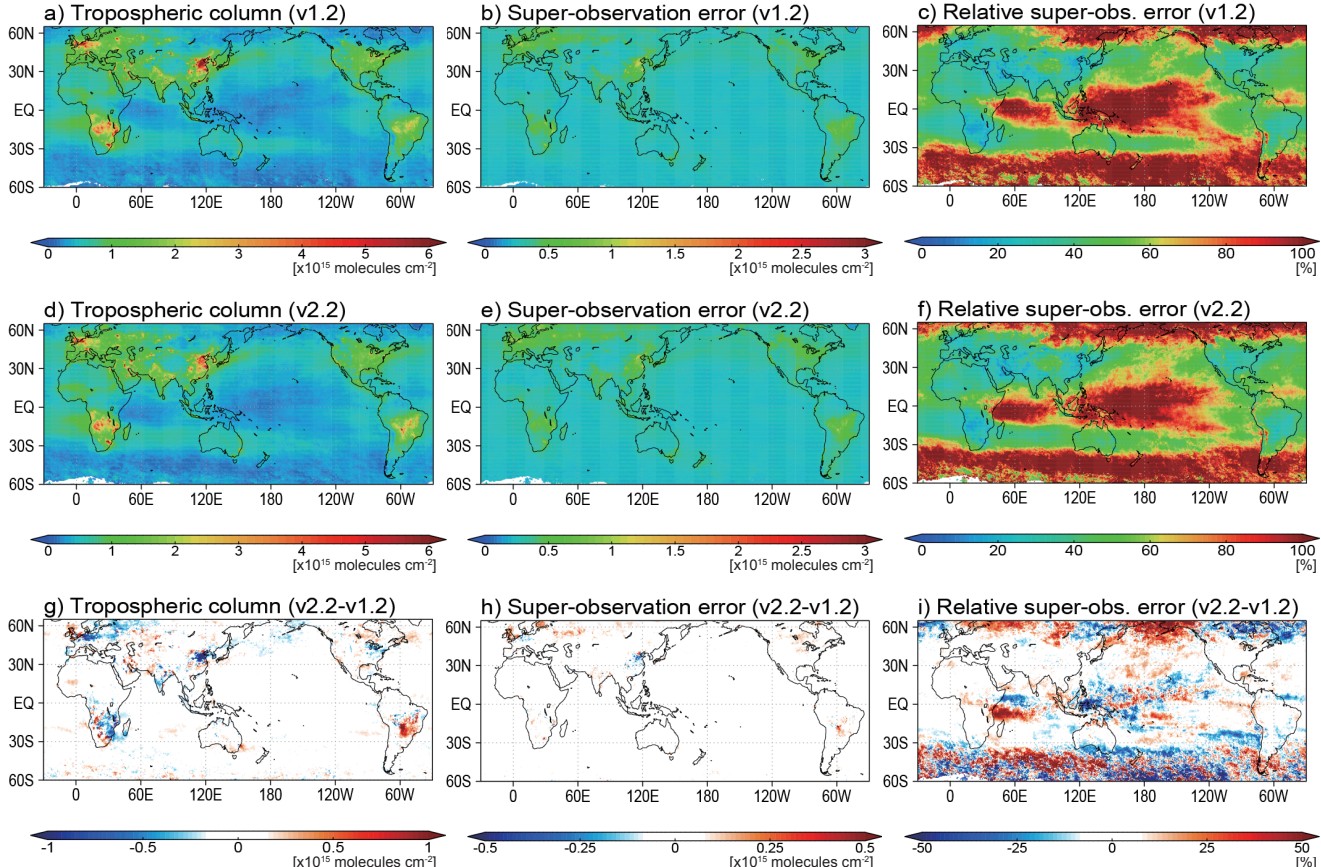

**Figure A1.** Global distribution of tropospheric NO$_2$ column (left), super-observation errors (middle), and relative super-observation errors (right) obtained from the Tropospheric Monitoring Instrument (TROPOMI) version 1.2 product during September 2018 (top) and TROPOMI version 2.2 product during September 2021 (middle), and the differences between TROPOMI versions 1.2 and 2.2 (bottom). The units of the tropospheric NO$_2$ column, super-observation errors, and relative super-observation errors are $\times 10^{15}$ molecules cm$^{-2}$, $\times 10^{15}$ molecules cm$^{-2}$, and percentage (%), respectively.