# Peer review of "A comparison of the impact of TROPOMI and OMI tropospheric $NO_2$ on global chemical data assimilation"

_Atmospheric Measurement Techniques, 2021_

## Author Comment (AC1)

A comparison of the impact of TROPOMI and OMI tropospheric NO2 on global chemical data assimilation: Reply to comments from anonymous referee #1

We would like to thank anonymous referee #1 for his or her careful reading and valuable comments, which have helped to significantly improve the manuscript. We have revised the manuscript corresponding to the referee's comments. Main changes we made are as follows:

1) Appendix A was added to discuss the seasonally varying bias of TROPOMI $NO_2$.

2) The discussion on the potential impacts of a low bias of TROPOMI in winter on the DA performance was added to Section 6.

3) The comparison of surface $NO_2$ concentrations derived from the control model simulation, TROPOMI DA, and OMI DA was added to Figure 9.

Individual comments (in black) and specific responses to them (in blue) are listed below. Texts (*Italicized font*) from the revised manuscript are in quotes.

To discuss the potential impacts of the TROPOMI retrieval algorithm updates, the submitted manuscript compared TROPOMI $NO_2$ version 1.2.2 product during September 2018 with version 2.2 product during September 2021 in Appendix A. At that time, the version 2.2 product was not available for April—May 2018. We noted that this comparison for different periods makes it difficult to distinguish the impacts of the algorithm updates from those of inter-annual changes. After the paper submission, the S5P-PAL reprocessing product became available for May 2018—July 2021. To provide more a consistent comparison for the same time period, we have revised Appendix B (corresponding to Appendix A of the submitted manuscript) to the comparison between version 1.2beta and S5P-PAL reprocessing products for May 2018. Although this change was not requested by the referees, we think that this update provides a better implication for the impacts of algorithm updates. The paragraph has been revised as follows:

(p. 41, l. 793—809)

"*In the latest version of the TROPOMI $NO_2$ product, the low bias compared to OMI QA4ECV is largely improved from the previous versions (van Geffen et al., 2021). To discuss the potential impacts of the retrieval algorithm updates on the DA performance, Figure B1 compares global distributions of tropospheric $NO_2$ column, super-observation errors, and relative super-observation errors (i.e., errors divided by concentrations) obtained from the TROPOMI version 1.2beta product, that was used in this study, and S5P-PAL reprocessing product (processed with same processor as version 2.3.1), that was released more recently, for May 2018. The S5P-PAL reprocessing product data were obtained from the S5P-PAL data portal (https://data-portal.s5p-pal.com). The algorithm updates from versions 1.2 to 2.3 led to increases in tropospheric $NO_2$ column amounts typically by 6% over polluted areas due to the algorithm updates from versions 1.2 to 2.3. These increases are mainly attributable to the improved*

*FRESCO cloud retrievals (van Geffen et al., 2021). In contrast, the relative super-observation errors over most regions except for the southern mid-latitudes are comparable between the products, with less than 0.2% differences in the mean relative super-observation error over 60°N–60°S. These differences are much smaller than the differences between the TROPOMI version 1.2beta and OMI QA4ECV products (by 19% in May 2018).*

*The improved TROPOMI retrievals would reduce the negative bias of the $NO_2$ concentration analysis compared to OMI and increase the estimated $NO_x$ emissions for areas with weak chemical non-linearity. The increase in $NO_x$ emissions would reduce the negative biases in ozone analysis under $NO_x$-limited ozone chemical regime. Meanwhile, the relative super-observation errors of TROPOMI retrievals were almost identical between versions 1.2beta and 2.3.1. This suggests that the DA efficiency, for example, to constrain detailed temporal and spatial variations, might not be largely affected by the algorithm updates."*

In this manuscript, the authors present a systematic comparison of the Tropospheric Monitoring Instrument (TROPOMI) version 1.2 and Ozone Monitoring Instrument (OMI) QA4ECV tropospheric NO2 column through global chemical data assimilation (DA) integration. The comparison of the impact of TROPOMI and OMI tropospheric NO2 on global chemical data assimilation is comprehensive. The topic of the manuscript fits the scope of AMT. The manuscript is mostly well written. However, some details of observation data and discussions are needed. The paper can be published after some minor revisions.

We appreciate careful reviews again.

The study is based on only two months (the period April–May 2018). To my knowledge, TROPOMI has strong negative bias in wintertime. If the study is conducted for the winter period or other months, will the conclusion be different? The discussion is missing in the paper.

Thank you for the important comments.

In the submitted manuscript, the following explanations were added to describe the potential impact of TROPOMI low biases compared to OMI on the validation results and top-down $NO_x$ emission estimates for April—May 2018 in the manuscript as follows:

(Section 3.3.2, p. 10, l. 306—307)

"*These results suggest that the results of TROPOMI DA were affected by the TROPOMI low bias compared to OMI, ...*"

(Section 4, p. 12, l. 358—359)

"*These differences reflect the low bias of TROPOMI retrievals compared to OMI retrievals.*"

(Section 4, p. 13, l. 381—383)

"*Overall, these results imply that top-down NOx emission estimates using TROPOMI version 1.2-1.3 products could be affected by the TROPOMI low biases compared to OMI, while ...*"

(Section 5.1, p. 14, l. 418—420)

"*Meanwhile, any biases in satellite NO2 retrievals will affect the surface ozone analysis. Surface ozone analysis biases are expected to be increased for a NOx-limited ozone chemical regime when using updated retrievals with reduced TROPOMI NO2 negative bias.*"

The main reasons for discussing the April—May 2018 period only are that (1) many aircraft-campaign observations are available and (2) there is an active ozone photochemical production during this time period. Meanwhile, as pointed out by the referee, the TROPOMI bias impact can be different in other seasons. Although it was not feasible to add a detailed evaluation for winter seasons in this study, we have added a figure and explanations to Appendix A and Section 6 to explore the potential impact of the larger TROPOMI biases in winter as follows:

(Section 6, p. 15, l. 465—472)

"*The systematic differences of TROPOMI version 1.2 compared to ground-based remote sensing and OMI are larger in winter than in other seasons over the polluted regions (Verhoelst et al., 2020; van Geffen et al., 2021; Lambert et al., 2021), consistent with Appendix A. The influence of negative biases related to the a-priori profile shape are mostly removed by using averaging kernels. However, because of the larger TROPOMI (version 1.2) negative bias compared to OMI in winter than in April-May, the relative DA performance between TROPOMI and OMI will depend on the season, especially over heavily polluted areas. Because of the availability of aircraft-campaign observational data for validation and the active photochemical production during the target period, this study focused on April–May 2018 only, and the impact of seasonally varying relative biases between OMI and TROPOMI has not been investigated.*"

(Section 6, p. 15, l. 476—482)

"*Lambert et al. (2021) and van Geffen et al. (2021) reported that the negative biases of the updated TROPOMI retrieval (versions 1.4.x and 2.x) compared to OMI are reduced to within 10%. Assuming a remaining bias of 10% compared to OMI, the improved TROPOMI retrievals would increase the estimated NOx emissions by 10–30% over Europe and eastern China in winter, compared to the DA using TROPOMI version 1.2beta. The increase in NOx emissions would reduce negative ozone biases*"

*in the DA analysis for a NOx-limited ozone chemical regime. Further investigations on the impacts of the seasonally varying retrieval biases would provide more detailed insights into the relative performance of TROPOMI and OMI DA.*"

(Appendix A, p. 40, l. 781—791)

"*As shown in Figure A1, the negative biases in TROPOMI tropospheric $NO_2$ column compared to OMI are larger in December 2018–February 2019 (by 25%, 19%, and 26% over Europe, the United States, and China, respectively) than in April–May 2018 (by 10%, 17%, and 16% over Europe, the United States, and China, respectively). In contrast, the differences in super-observation errors between TROPOMI and OMI are relatively constant over time. The differences in the relative super-observation errors (i.e., errors divided by concentrations) obtained from TROPOMI and OMI are smaller in winter than in other seasons over Europe and China because of the larger bias of the TROPOMI tropospheric $NO_2$ column compared to OMI in winter than in other seasons.*

*   The strong negative biases in TROPOMI retrievals in winter would increase the negative bias in NO2 concentration analysis and reduce the estimated NOx emissions. Meanwhile, these differences in relative super-observation errors of TROPOMI retrievals between winter and other seasons suggests that TROPOMI DA might provide less constraints on spatial and temporal variations in $NO_2$ even in winter than other seasons, and would still better constraints than OMI DA.*"

As discussed above and in the revised manuscript carefully, the TROPOMI (version 1.2beta) negative bias compared to OMI has been greatly reduced in the latest reprocessed product (version 2.3.1) that was released in December after this work was conducted. In addition, aircraft-campaign validation data are limited, and the ozone photochemical production is inactive in winter. Therefore, while discussing its potential impact in the revised manuscript, we think that it is not essential to add a detailed evaluation result on impacts of the wintertime negative bias on DA using the TROPOMI version 1.2.x product in this study.

Specific comments:

L71: typo: rfaction to fraction.

Modified

L88-93 section 2.2.1, can you please provide more details about the NO2 observations in the Atom aircraft-campaign? Such as: what is the time window of the no2 observations on each day? The frequency of the no2 observations, per minute? Per hour?

We added the description on NO$_2$ measurements in ATom aircraft-campaign including time window, as follows:

(p. 4, l. 91—94)

"*The NO$_2$ concentrations were measured via the NOAA NOyO3 4-channel chemiluminescence instrument per 1 second with precision of 5–10 pptv (https://espoarchive.nasa.gov/instrument/ NOyO3). The merged dataset of flight data with 10-second means was used for the validation.*"

L169: spatial representativeness error, should it be $\sqrt{\sigma_m^2 + \sigma_r^2}$?

This part was modified.

L 265-261: How did you compare the vertical profiles between aircraft measurements to the model simulation? Can you give more details? Did you average the profiles over the area?

More detailed descriptions on comparison method and area definition were added.

(p. 9, l. 268—271)

"*... over coastal areas of the western United States (117.25–122.5°W, 32–37°N). At first the control model simulation and data assimilation results were sampled at observation locations, and then the observation data, the control model simulation, and the data assimilation were averaged on each day over the coastal areas of the western United State.*"

L417-450: The study time period of the data assimilation is April-May not the whole year. Please mention this in the conclusion.   If you get the same conclusions or not when you include winter period. It could be nice that you can add some discussion here.

Please see our reply above.

L426: It is not accurate to conclude the global change of NOx emissions per year. Please rephrase the sentence or mention the time period.

We mentioned the time period for this study:

(p. 15, l. 448)

"*Global total NO$_x$ emission for April 15–May 31 2018 was increased ...*"

---

## Author Comment (AC2)

A comparison of the impact of TROPOMI and OMI tropospheric NO2 on global chemical data assimilation: Reply to comments from anonymous referee #2

We would like to thank anonymous referee #2 for his or her careful reading and valuable comments, which have helped to significantly improve the manuscript. We have revised the manuscript corresponding to the referee's comments. Main changes we made are as follows:

1) Appendix A was added to discuss the seasonally varying bias of TROPOMI $NO_2$.

2) The discussion on the potential impacts of a low bias of TROPOMI in winter on the DA performance was added to Section 6.

3) The comparison of surface $NO_2$ concentrations derived from the control model simulation, TROPOMI DA, and OMI DA was added to Figure 9.

Individual comments (in black) and specific responses to them (in blue) are listed below. Texts (*Italicized font*) from the revised manuscript are in quotes.

At the time of the paper submission, the version 2.2 product was not available for April—May 2018, the submitted manuscript compared TROPOMI $NO_2$ version 1.2.2 product during September 2018 with version 2.2 product during September 2021 in Appendix A. This comparison for different periods caused a limitation to clearly distinguish the impacts of the algorithm updates from those of inter-annual changes. After the paper submission, the S5P-PAL reprocessing product became available for May 2018—July 2021. To provide more a consistent comparison for the same time period, we have revised Appendix B (corresponding to Appendix A of the submitted manuscript) to the comparison between version 1.2beta and S5P-PAL reprocessing products for May 2018. Although this change was not requested by the referees, we think that this update provides a better implication for the impacts of algorithm updates. The paragraph has been revised as follows:

(p. 41, l. 793—809)

"*In the latest version of the TROPOMI $NO_2$ product, the low bias compared to OMI QA4ECV is largely improved from the previous versions (van Geffen et al., 2021). To discuss the potential impacts of the retrieval algorithm updates on the DA performance, Figure B1 compares global distributions of tropospheric $NO_2$ column, super-observation errors, and relative super-observation errors (i.e., errors divided by concentrations) obtained from the TROPOMI version 1.2beta product, that was used in this study, and S5P-PAL reprocessing product (processed with same processor as version 2.3.1), that was released more recently, for May 2018. The S5P-PAL reprocessing product data were obtained from the S5P-PAL data portal (https://data-portal.s5p-pal.com). The algorithm updates from versions 1.2 to 2.3 led to increases in tropospheric $NO_2$ column amounts typically by 6% over polluted areas due to the algorithm updates from versions 1.2 to 2.3. These increases are mainly attributable to the improved FRESCO cloud retrievals (van Geffen et al., 2021). In contrast, the relative super-observation errors*

*over most regions except for the southern mid-latitudes are comparable between the products, with less than 0.2% differences in the mean relative super-observation error over 60°N–60°S. These differences are much smaller than the differences between the TROPOMI version 1.2beta and OMI QA4ECV products (by 19% in May 2018).*

*The improved TROPOMI retrievals would reduce the negative bias in $NO_2$ concentration analysis and increase the estimated $NO_x$ emissions for areas with weak chemical non-linearity. The increase in $NO_x$ emissions would reduce the negative biases in ozone analysis under $NO_x$-limited ozone chemical regime. Meanwhile, the relative super-observation errors of TROPOMI retrievals were almost identical between versions 1.2beta and 2.3.1. This suggests that the DA efficiency, for example, to constrain detailed temporal and spatial variations, might not be largely affected by the algorithm updates.*"

The manuscript by Sekiya et al. compared the global chemical data assimilation results when using NO2 retrievals from TROPOMI and OMI. The TROPOMI posterior NO2 shows better agreement with NO2 observations and smaller magnitude than the OMI one. The manuscript is generally well-written. The topic fit the scope of AMT. The result is important in interpreting existing NOx data assimilations. I suggest publication after addressing the following comments.

We appreciate in careful reading and comments again.

L7, if TROPOMI NO2 is biased generally low, would the comparison with independent data improved for the wrong reason?

Negative biases in $NO_2$ concentration analysis against independent observations were increased by TROPOMI DA for some cases. Meanwhile, the agreements with independent observations were improved by better constraints on spatial and temporal variations in $NO_2$ in TROPOMI DA than in OMI DA, because of lower super-observation errors in TROPOMI than those in OMI. We added description to abstract as follows:
(p. 1, l. 9)
"*... because of better capturing spatial and temporal variability by TROPOMI DA.*"

Figure 1. Please provide the resolution these data are gridded to in the figure description.

We added the following sentence on mapping grids to caption of Figure 1:
(p. 26)
"*The values are mapped onto 0.56° resolution grids.*"

How much do precision error and the number of observations in the super-observation grid each contribute to the smaller super-observation errors in TROPOMI data?

The precision error improvements were more important than the increases in the number of observations per a grid. Meanwhile, the relative contributions of precision error improvements were smaller over polluted regions than over remote regions. The following description was added:
(p. 7, l. 208—209)
"*The improved S/N ratio and stripes contributed to about 80% and almost 100% of smaller super-observation errors over polluted and remote regions, respectively*"

Line 221-222, it would be clearer to first explain what the range of chi-square is, and what do values larger and smaller than 1 generally mean.

We have revised the explanation on meaning of chi-square values as follows and moved it to before results.
(p. 8, l. 226—228)
"*$\chi^2$ value is used to diagnose balance between actual errors and estimated errors. When $\chi^2$ value is larger (smaller) than the ideal value of 1, it is suggested underestimated (overestimated) background error covariance or observation errors.*"

The range of chi-square values (i.e., standard deviation) were also added:
(p. 8, l. 232—233)
"*The mean values of estimated $\chi^2$ with standard deviation range … was 0.99±0.25 for TROPOMI DA, whereas the mean $\chi^2$ of 1.17±0.19 for OMI DA is …*"

Figure 2, I am a bit surprised that a large portion of the TROPOMI DA improvement is over the ocean, where there is no emissions. Please explain what possibly causes this.

The DA system used in this study optimizes $NO_2$ concentrations and NOx emissions simultaneously, which led to the improvements over land and ocean where there is no emissions. The variables which are optimized by DA is emphasized in Section 2.3.2 as follows:
(p. 5, l. 139—140)
"*…, which optimizes ozone and related chemical species' concentrations, and ozone precursors' emissions simultaneously.*"

We also added the explanation about improvements in TROPOMI DA over oceans as follows:

(p. 9, l. 249—251)

*"Over the oceans in the tropics and midlatitudes, higher vertical sensitivity (i.e., averaging kernels) in TROPOMI than OMI in the lower troposphere and above clouds contributed to the improved performance, through ship and lightning NOx emission corrections and direct NO₂ concentration modifications."*

L250-251, I am confused about the "regardless of the TROPOMI low bias" part. Is this only true because you calculate RMSE against the TROPOMI observations?

Yes, because the RMSE was estimated against the assimilated TROPOMI measurement, it is not affected by the TROPOMI low bias. This part was modified as follows:

(p. 9, l. 263—264)

*"... the DA efficiency by TROPOMI was evaluated based on RMSE against assimilated observation Itself. It is determined by the amount and quality of TROPOMI data, regardless of the TROPOMI low bias."*

Figure 4, Please provide more information on what is being optimized in the DA. Are both NO2 concentrations and emissions optimized at the same time?

The DA system used in this study optimizes NO₂ concentrations and NOₓ emissions at the same time. We add this explanation to Section 2.3.2 (please see our reply above).

Are emissions all in the surface layer? If not, how are they distributed vertically, and how does DA adjust emissions differently at different layers?

Anthropogenic (except for aviation), biomass burning, and soil emissions are in the lowest model layer. The following sentence was added:

(p. 5, l. 133—134)

*"These emissions are released at the lowest model layers."*

Lightning NOx sources are vertically distributed using the C-shaped profile given by Pickering et al. (1998). The following description was added:

(p. 5, l. 134—136)

*"... the parameterization proposed by Price and Rind (1992), with the assumption for vertical distribution of lighting NOx source based on the C-shaped profile given by Pickering et al. (1998)."*

Data assimilation adjusts 3-D multiplication factors for the lightning NO production rate. The following sentence was added:

(p. 6, l. 155—157)

"*For lightning NO$_x$, multiplication factors for the lightning NO production rate were adjusted differently at different model layers using the method proposed by Miyazaki et al. (2014) and the background error covariance matrix.*"

L287, a similar comment as a previous one, if there are systematic low biases in TROPOMI data, why do its DA results have better agreement with independent data?

As mentioned in the second reply, TROPOMI version 1.2 data have negative biases, which led to increased negative biases in TROPOMI DA against independent surface in-situ observations for some cases. Meanwhile, TROPOMI DA better captured spatial and temporal variations mainly because of reduced TROPOMI super-observation errors associated with improved S/N ratio, increased number of observation data per a grid, and TROPOMI stripes. This part was modified to explain cause of the improvements as follows:

(p. 10, l. 302)

"*... reduced RMSE by 23% because of better capturing spatial and temporal variations, but increased ...*"

Also, we added the following sentence at the end of this paragraph to emphasis how the performance TROPOMI DA is improved.

(p. 10, l. 307—308)

"*..., while TROPOMI DA provided better constraints on spatial and temporal variations in NO$_2$ concentrations than OMI DA.*"

L330, could you also add a figure showing the changes in NO2 concentrations from the two DA?

We added global maps of the surface NO$_2$ concentration analysis derived from TROPOMI DA and the differences from the control model simulation and OMI DA to Figure 9. The corresponding description was also changed:

(p. 12, l. 357—358)

"*..., which led to smaller surface NO2 concentrations (Figure 9). These ...*"

L339-351, based on the low biases in TROPOMI NO2 retrievals and the comparisons here, what is

the implication for existing DA and inversion results using this version of TROPOMI NO2?

We added the implication for DA using TROPOMI NO2 version 1.2-1.3:
(p. 13, l. 381—384)
"*Overall, these results imply that top-down NOx emission estimates using TROPOMI version 1.2-1.3 products could be affected by the TROPOMI low biases compared to OMI, while top-down estimates using TROPOMI have the potential for constraints on detailed spatial and temporal variations based on validation results (c.f., Section 3.3).*"

L446-447, would you expect the low biases in TROPOMI NOx emissions reduce using this new product, and by how much?

Based on the complemental analysis presented in Appendix B in the revised manuscript, we expect to reduce impacts of TROPOMI low bias on emission estimates if the new version products are used. This part was modified to be more quantitative, as follows:
(p. 16, l. 474—479)
"*These new versions largely remove the bias with respect to the OMI 475 QA4ECV product for all seasons, especially in winter over polluted areas (van Geffen et al., 2021). Lambert et al. (2021) and van Geffen et al. (2021) reported that the negative biases of the updated TROPOMI retrievals (versions 1.4.x and 2.x) compared to OMI are reduced to within 10%. Assuming a remaining bias of 10% compared to OMI, the improved TROPOMI retrievals would increase the estimated NOx emissions by 10–30% over Europe and eastern China in winter for areas with a weak chemical non-linearity, compared to the DA using TROPOMI version 1.2beta.*"

Reference
Pickering, K. E., Wang, Y., Tao, W.-K., Price, C., and Müller, J.-F.: Vertical distributions of lightning NOx for use in regional and global chemical transport models, J. Geophys. Res., 103, 31 203–31 216, https://doi.org/10.1029/98JD02651, 1998